# Design, Synthesis and Biological Evaluation of Neocryptolepine Derivatives as Potential Anti-Gastric Cancer Agents

**DOI:** 10.3390/ijms231911924

**Published:** 2022-10-07

**Authors:** Yunhao Ma, Yanan Tian, Zhongkun Zhou, Shude Chen, Kangjia Du, Hao Zhang, Xinrong Jiang, Juan Lu, Yuqing Niu, Lixue Tu, Jie Wang, Huanxiang Liu, Hongmei Zhu, Peng Chen, Yingqian Liu

**Affiliations:** 1School of Pharmacy, Lanzhou University, Lanzhou 730000, China; 2The Second Clinical Medical College, Lanzhou University, Lanzhou 730000, China

**Keywords:** neocryptolepine derivatives, AKT, AGS cell, HGC27 cell, PI3K/AKT signaling pathway

## Abstract

Natural products play an important role in drug development and lead compound synthesis. Neocryptolepine is a polycyclic quinoline compound isolated from *Cryptolepis sanguinolent*. The cytotoxicity of neocryptolepine to gastric cancer cells AGS, MKN45, HGC27, and SGC7901 was not very strong, and it also had certain toxicity to gastric mucosa cells GES-1. Therefore, a series of neocryptolepine derivatives were synthesized by the modification of the structure of neocryptolepine, and their cytotoxicity was evaluated. The results showed that compounds **C5** and **C8** exhibited strong cytotoxicity to AGS cells. The cell colony formation and cell migration experiments suggested that compounds **C5** and **C8** could inhibit the proliferation and cell migration of AGS and HGC27 cells. Cell cycle and apoptosis experiments showed that compounds **C5** and **C8** did not cause the apoptosis of AGS and HGC27 cells but, mainly, caused cell necrosis. Compound **C5** had no significant effect on AGS and HGC27 cell cycles at low concentration. After treatment with AGS cells for 24 h at high concentration, compound **C5** could significantly arrest the AGS cell cycle in the G2/M phase. Compound **C8** had no significant effect on the AGS and HGC27 cell cycles. The results of molecular docking and Western blot showed that compounds **C5** and **C8** might induce cytotoxicity through the PI3K/AKT signaling pathway. Therefore, compounds **C5** and **C8** may be promising lead compounds for the treatment of gastric cancer.

## 1. Introduction

Cancer is the second leading cause of human death after cardiovascular disease, and cancer of the digestive system accounts for about 50% of all cancers [1]. According to the global cancer statistics in 2018, gastric cancer is the most common cancer of digestive system tumors, with about 1.03 million new cases of gastric cancer worldwide, ranking fifth in the incidence of malignant tumors and becoming the third leading cause of cancer death [2]. The five-year survival rate of gastric cancer is less than 25%, and we are often powerless for patients with advanced gastric cancer [3,4]. The methods of cancer treatment mainly include radiotherapy, chemotherapy, surgery, and gene therapy, but chemotherapy is a necessary means to treat solid tumors at present. Compared with other cancer treatments, oral chemotherapy drugs have the advantages of low cost and strong patient compliance. However, chemical drugs have large side effects, and drug resistance is difficult to solve [5,6]. Therefore, in order to overcome these obstacles, it is very necessary to develop new and less toxic chemical drugs to treat gastric cancer.

Among natural products, alkaloids are one of the main natural products. Alkaloids were discovered and used as early as 4000 years ago, and alkaloids and their derivatives have been used as drug sources to treat various diseases around the world, including the development of anticancer drugs [7]. Traditional alkaloids extracted from plants have played a huge role in the past [8], and more than 5000 alkaloids have been reported since the discovery of the first alkaloid, morphine, in 1805 [9]. A large number of studies have also shown that alkaloids performed excellent cytotoxicity to different cancers, including human melanoma, breast cancer, pancreatic cancer, colorectal cancer, oral cancer, liver cancer, and gastric cancer [10,11,12,13,14,15]. *Cryptolepis sanguinolenta* is a vine that grows in some African countries, and the roots of this plant have proven to be a rich source of indoline-quinoline alkaloids [16]. In recent years, neocryptolepine, a promising natural quinoline indole alkaloid, has attracted much attention. Some neocryptolepine derivatives have strong cytotoxicity to leukemia cells MV4-11, with an IC_50_ of 42 nM and also to lung cancer cells with an IC_50_ of 197 nM [17]. Neocryptolepine and its derivatives have a wide range of biological activities, and compounds containing this ring system have antifungal, antibacterial, antiviral, and cytotoxic activity [18,19,20,21]. In fact, indolequinoline alkaloids have good anticancer activities, and semi-synthetic analogues of these neocryptolepine can be prepared, which have shown great potential effects of cytotoxic agents [22]. Therefore, indolequinoline alkaloids are considered as a promising framework for drug development and can be further developed as effective anticancer drugs [8,22].

Due to the complex structure of natural products, different chemical components have different anticancer mechanisms [23]. Studies have shown that some alkaloids can induce apoptosis and cell cycle arrest [24]. The activation of cancer signaling pathways is common in the occurrence of cancer [25]. Multiple signaling pathways are involved in the occurrence of gastric cancer [26]. Phosphatidylinositol 3-kinase (PI3K)/protein kinase B (AKT) is an important signaling pathway in cells. When the PI3K/AKT signaling pathway is abnormally activated, it may cause the activation of downstream signaling molecules, thus affecting the development of gastric cancer, lung cancer, and other malignant tumors [27,28,29]. Studies have shown that the development of gastric cancer is related to excessive cell proliferation and inhibition of apoptosis, and activation of the PI3K/AKT cell signaling pathway often prevents programmed cell death [30]. The PI3K/AKT cell signaling pathway plays an important role not only in tumor development but also in tumor therapy, and many new targeted agents are realized by acting on relevant targets of the PI3K/AKT signaling pathway [31,32].

In the previous studies, in addition to numerous activity tests and mechanistic studies on the parent structure of neocryptolepine, a great deal of work has been done on the derivatives with the **C11** position substitution of neocryptolepine. Some studies have found that neocryptolepine derivatives have good antibacterial, anti-proliferative, and antifungal activities [33,34,35]. For example, in 2009, Ibrahim El Sayed et al. introduced a long amino-alkyl chain substitution at the **C11** position of neocryptolepine. A series of derivatives were prepared and further tested for their inhibitory activity against Plasmodium. Among them, the IC_50_ of the most active compound was 0.043 μM [36]. In 2012, Li Wang et al. reported the effects of derivatives obtained by modifying the **C11** position with a variety of amino alkyl chains on the human leukemia MV4-11 cell line in anti-proliferation experiments. The experimental results showed that most of the derived molecules had good anti-proliferation activity, but they also had high cytotoxicity [35,37]. Therefore, after performing the cytotoxicity study of 8-chloroneocryptolepine, the substitution at **C11** might be ideal to improve the inhibitory activity of 8-chloroneocryptolepine on cancer cells.

As a result, we aimed to perform the modification of the **C11** position of 8-chloroneocryptolepine, and a series of neocryptolepine derivatives were synthesized. The cytotoxic effects of neocryptolepine derivatives on liver cancer SMMC7721 and gastric cancer AGS cells were evaluated in vitro. The results showed that compounds **C5** and **C8** exhibited strong cytotoxicity against gastric cancer cells and may be promising lead compounds in the treatment of gastric cancer.

## 2. Results and Discussion

### 2.1. Chemistry

As shown in Figure 1, in previous work, the structure of neocryptolepine was optimized by structural modification, and 8-chloroneocryptolepine performed good anti-fungal effects. The active functional group piecing strategy has been widely used in the field of derivative synthesis and structure optimization of antitumor compounds [24]. Moreover, the introduction of amino long-chain alkanes at the **C11** position of neocryptolepine could improve the cytotoxic effect of the compound [36,37]. Therefore, the cytotoxicity evaluation of a series of derivatives, by introducing an active functional group to **C11** of 8-chloroneocryptolepine, is a promising strategy for the development of a lead compound with anti-tumor drugs.

The synthesis of neocryptolepine derivatives and intermediates was shown in Figure 2. Intermediate I was easily obtained with a yield of more than 80%. Indoles, trichloroacetyl chloride, and tetrahydrofuran were acylated to obtain intermediate I. Subsequently, intermediate II was obtained with the reaction of intermediate I and *N*-methylaniline, and the yield was more than 60%. Intermediate III was obtained by the reaction of intermediate II with diphenyl ether, and intermediate IV was obtained by the interaction with phosphorus oxychloride, with a yield of more than 60%. Intermediate IV reacted with the corresponding alcohol hydroxyl compound in N, N-dimethylformamide (DMF) to obtain the corresponding compounds **A1–A10**, and with the corresponding phenylhydrazine compound to obtain compounds **B1–B9**, with the corresponding amide compound to obtain compounds **C1–C10**, **D1–D6**. The structures of these compounds can be found in Table 1. It is important to note that the commercially available raw materials were obtained through the synthesis of intermediates and final products with a good yield. The structures of the target compounds were confirmed by ^1^H and ^13^C NMR and MS.

### 2.2. Cytotoxic Activity In Vitro and Structure-Activity Relationship (SAR)

The cytotoxicity of intermediate IV and neocryptolepine derivatives on gastric cancer AGS cells and hepatoma SMMC7721 cells was determined by MTT assay. Based on our results of cytotoxicity, the structure–activity relationship was studied for neocryptolepine derivatives (as shown in Figure 1). Firstly, the **C11** position of 8-chloroneocryptolepine was substituted with ether groups. However, in Table 2, the results suggest that the IC_50_ of compounds **A1–A10** was greater than 50 μM for SMMC7721, but some of the compounds had a strong cytotoxicity on AGS cells. According to the cytotoxicity results, the *para*-site substitution of F atom (**A4**) is better than the *meta*-site (**A3**) and *ortho*-site (**A2**) substitution, and the methoxy *ortho*-site (**A5**) substitution is more cytotoxic than the *meta* (**A6**) and *para*-site (**A7**) substitution, as well as the dimethoxy (**A8**) substitution, on benzene ring.

In 2016, Masashi Okada et al. performed an anti-proliferative activity assay on breast cancer MDA-MB-453 cells, colorectal cancer WiDr Cells, and ovarian cancer SKOv3 cells by introducing amino long-chain alkanes at the **C11** position. The experimental results showed that the introduction of amino long-chain alkanes at the **C11** position was beneficial to the improvement of anti-proliferative activity [38]. Based on the above structural modification and cytotoxicity, the **C11** position of 8-chloroneocryptolepine was substituted by hydrazine group to obtain compounds **B1–B9**. According to the cytotoxicity results in Table 3, the substitution of hydrazine group (**B1–B9**) was more cytotoxic than that of ether group (**A1–A10**). The results show that, in hydrazine group substitution, the substitution of F or methyl group increased its activity. Moreover, mono-substituted (**B2–B4**) F atom on benzene rings was more cytotoxic than disubstituted (**B5**) F atoms, and *meta*-substituted (**B3**) F atom was more cytotoxic than *para*-substituted (**B4**) F atom. The substitution of *ortho* (**B6**) methyl groups on benzene rings was better than that of *meta* (**B7**) and *para* (**B8**) groups.

Compounds **C1–C10** were synthesized by substituting **C11** sites with amide groups through electron iso-arrangement. In Table 4, by comparison and verification of experimental results, compounds **C5** and **C8** were found to be more active against AGS cells, and their possible mechanisms were studied at the cellular level. Moreover, as shown in Table 5, the cytotoxic effects of compounds **C5** and **C8** were significantly better than the positive drug cisplatin and the parent nucleus 8-chloroneocryptolepine. According to the cytotoxicity results in Table 4, the *para*-F (**C5**) atomic substitution of benzene ring was more cytotoxic than the *ortho* (**C3**) and *meta*-position (**C4**), and the *meta*-position (**C7**) and *para*-methoxy (**C8**) substitution were better than the *ortho*-substitution (**C6**). The increase in alkyl side chain (**C9** and **C10**) showed better cytotoxicity against AGS cells than compounds **A1–A10**. The structure–activity relationship of compounds **C1–C10** can also be found in Figure 1.

Further evaluation of the cytotoxicity of 8-chloroneocryptolepine on gastric cancer AGS and liver cancer SMMC7721 cells revealed that its cytotoxicity was not ideal (as shown in Table 5). Finally, the **C11** position of 8-chloroneocryptolepine was substituted with a sulfonamide group to obtain compounds **D1–D6**. However, in Table 5, the cytotoxicity results suggested that the cytotoxicity for SMMC7721 cells was very weak. Therefore, the substitution of the sulfonamide group at the **C11** position was not an ideal choice. However, in the cytotoxicity results of AGS cells, the *meta*-positional (**D2**) substitution of F atom was less potent (or less cytotoxic) than *ortho*- (**D1**) and *para*-substitutions (**D3**). The *ortho*-methyl substitution (**D4**) in benzene ring was more cytotoxic than the *para*-methyl (**D5**) substitution.

### 2.3. Preliminary Cytotoxic Mechanism of Compounds **C5** and **C8** against AGS and HGC27 Cells

#### 2.3.1. The Cytotoxic Effect of Compounds **C5** and **C8** on Gastric Cancer Cells

The cytotoxicity of compounds **C5** and **C8** were systematically evaluated by an MTT assay. As shown in Figure 2, we performed cytotoxicity experiments on five gastric cancer cell lines—AGS, HGC27, MKN45, MGC803, and SGC7901—using neocryptolepine and cisplatin as the control. The results showed that the IC_50_ values of neocryptolepine were 20, 18, 19, 40, and 37 μM on AGS, HGC27, MKN45, MGC803, and SGC7901 cells, after 48 h, respectively. Compared with the cytotoxicity of the parent nucleus of neocryptolepine, we found that compounds **C5** and **C8** had stronger cytotoxicity by structural modification. The IC_50_ values of compound **C5** on AGS, HGC27, MKN45, MGC803, and SGC7901 cells, for 48 h, were 9.2, 6.6, 5.9, 13, and 8.7 μM, respectively. The IC_50_ values of compound **C8** on AGS, HGC27, MKN45, MGC803, and SGC7901 cells, after 48 h treatment, were 6.9, 4.3, 3.5, 10, and 10 μM, respectively. Compared with the positive drugs, compounds **C5** and **C8** showed significantly stronger cytotoxicity than cisplatin. In Figure 2G, it was found that the cytotoxic effects of compounds **C5** and **C8** to normal cells (IC_50_ = 12.8 and 12.6 μM, respectively) were relatively weaker than gastric cancer cells. We also performed concentration-dependent and time-dependent experiments on the cytotoxicity of compound **C8** and cisplatin to AGS, and the results showed (Figure 2) that compound **C8** and cisplatin could inhibit the growth of gastric cancer cell lines AGS, HGC27, and MKN45 in a concentration-dependent and time-dependent manner. Moreover, compound **C8** showed strong cytotoxicity to gastric cancer cells at 5 μM, while cisplatin was weak. Therefore, the results of the MTT assay showed that we improved the cytotoxicity of the parent nuclear structure of neocryptolepine by structural modification, and compounds **C5** and **C8** showed stronger cytotoxicity effects compared with the positive drug cisplatin.

#### 2.3.2. Compounds **C5** and **C8** Inhibited the Proliferation and Migration of AGS and HGC27 Cells

To study the effects of compounds **C5** and **C8,** on the proliferation of AGS and HGC27 cells, the colony formation assays were performed. The results showed that, as shown in Figure 3A, compound **C8** at 1 μM and 2 μM, as well as compound **C5** at 2 μM and 4 μM, could inhibit the proliferation of AGS HGC27 cells. When the concentration of compound **C8** was 4 μM and compound **C5 was** at 6 μM, the proliferation of AGS and HGC27 cells was completely inhibited, and no cell clones were formed. Therefore, our results showed that compounds **C5** and **C8** had a certain inhibitory effect on the proliferation of AGS and HGC27 cells, and the proliferation of AGS and HGC27 cells could be completely inhibited when compounds **C5** and **C8** reached a certain concentration.

The migration of tumor cells is an important embodiment of the lethal effect of tumor. Therefore, cell migration experiments were performed to study the effects of compounds **C5** and **C8** on the migration ability of AGS and HGC27 cells. Results are shown in Figure 3C, after treatment with compound **C8** at 1.25 μM for 48 h, and the migrating ability of AGS and HGC27 cells was significantly inhibited. As the concentration of compound **C8** increased, the number of AGS cells decreased from 243 to 28, and the number of HGC27 cells decreased from 170 to 47. Similarly, compound **C5** inhibited the migration of HGC27 cells in a concentration-dependent manner. However, our results showed that compound **C5** did not have an obvious effect on the migration of AGS cells. It may be that the tumor specificity caused the different inhibitory effect of compounds on AGS and HGC27 cells. In conclusion, different concentrations of compound **C8** inhibited the migration of AGS and HGC27 cells, and compound **C5** also inhibited the migration of HGC27 cells at certain concentrations.

#### 2.3.3. The Effects of Compounds **C5** and **C8** on AGS and HGC27 Cell Cycle

Cell cycle regulation plays an important role in anti-tumor drugs. Therefore, to evaluate the effect of neocryptolepine derivatives on cell cycle, flow cytometry was used to test the changes of AGS and HGC27 cell cycle after neocryptolepine derivatives treatment. Results are shown in Figure 4A, AGS cells were treated with 2.5 μM and 5 μM compound **C8** for 24 h, and there was no significant change in AGS and HGC27 cell cycles. After the AGS and HGC27 cells were treated with different concentrations of compound **C5** for 24 h, it was found that HGC27 cells died when treated with 10 μM of compound **C5**. Therefore, we reduced the concentration of compound **C5**. The results showed that compound **C5** had no significant change in the HGC27 cell cycle when treated with 2.5 μM and 5 μM. However, the AGS cells were treated with 10 μM compound **C5** for 24 h, the AGS cells were mainly arrested in the G2/M phase. In the apoptosis experiment, although compound **C5** caused the necrosis of most AGS cells after treatment for 48 h, AGS cells did not die completely after treatment with compound **C5** for 24 h in the cell cycle. The cell cycle experiments suggested that compounds **C5** and **C8** had no significant effect on the cell cycles of AGS and HGC27 cells at low concentration, while compound **C5** could significantly block AGS cells in G2/M phase at high concentration.

#### 2.3.4. The Effects of Compounds **C5** and **C8** on Apoptosis of AGS and HGC27 Cells

Apoptosis is a common characteristic of most chemotherapeutic drugs that can exert antitumor effects. To investigate whether the neocryptolepine derivatives exert cytotoxic effects by cell apoptosis, we detected the effects of compounds **C5** and **C8** on the apoptosis of AGS and HGC27 cells by flow cytometry. The results (Figure 4C) showed that, when the concentration of compound **C8** was 2.5 μM, it did not cause the apoptosis or necrosis of AGS and HGC27 cells, but when the concentration reached 5 μM, compound **C8** had caused the necrosis of AGS and HGC27 cells, and when the concentration reached 10 μM, all AGS and HGC27 cells had apoptosis or necrosis. Similarly, compound **C5** showed the same effect in AGS and HGC27 cells, but it should be noted that, after treatment with 10 μM compound **C5** for 48 h, most AGS cells died. Moreover, cell cycle results also showed that compound **C5** treatment of AGS cells, at a concentration of 10 μM for 24 h, did not completely cause the death of AGS. Therefore, it can be concluded that compound **C5** and **C8** do not exert cytotoxic effects, mainly, through apoptosis but directly lead to cell necrosis.

#### 2.3.5. Molecular Docking of Compounds **C5** and **C8** with AKT Protein

The activation of carcinogenic signaling pathway is very frequent in the occurrence of cancer, and multiple signaling pathways are involved in the development of gastric cancer [25,26]. To find the cause of cytotoxicity of neocryptolepine derivatives to gastric cancer cell lines, that is, the intracellular targets of compounds **C5** and **C8**, a series of proteins related to cancer cell signaling pathways were docked with compounds **C5** and **C8** by molecular docking experiments. The results (Figure 5) showed that both compounds **C5** and **C8** had the best scores with the AKT protein. Its scores with the AKT protein were −8.529 and −8.359 kcal/mol, respectively. The side of the benzene ring in the small molecule is close to the hydrophobic region consisting of LEU210, TYR263, LEU264, VAL270, and TYR272, forming hydrophobic interactions with the receptor. The benzene ring of the small molecule forms a Π–Π interaction with the pyrrole of tryptophan 80. Therefore, the molecular docking results suggested that AKT may be a potential intracellular target of compounds **C5** and **C8**.

#### 2.3.6. Compounds **C5** and **C8** Inhibited AGS and HGC27 Cell Growth by Regulating PI3K/AKT Signaling Pathway

A variety of cell signaling pathways are involved in the occurrence of gastric cancer [26]. PI3K/AKT is an important signaling pathway, and when this pathway is abnormally activated, it affects the expression of downstream related proteins, thus affecting the occurrence and development of gastric cancer [39]. To further verify the results of molecular docking, Western blot experiments were performed to study the effects of compounds **C5** and **C8** on PI3K/AKT signaling pathway-related proteins in AGS and HGC27 cells. Results are shown in Figure 6, after AGS and HGC27 cells were treated with 5 μM compounds **C5** and **C8** for 48 h, and the expression levels of PI3KCA, AKT, and p-Akt proteins in the PI3K/AKT signaling pathway were down-regulated. The results further confirmed the molecular docking results: compounds **C5** and **C8** may act as AKT inhibitors to play an anti-proliferation role in gastric cancer. In conclusion, compounds **C5** and **C8** may inhibit the proliferation of gastric cancer AGS and HGC27 cells through the PI3K/AKT signaling pathway.

## 3. Materials and Methods

### 3.1. Reagents

All chemical reagents and solvents were of reagent grade or purified according to standard methods before use. The ^1^H NMR and ^13^C NMR spectra were recorded using JNM-ECS-400 MHz and 100 MHz. Mass spectrometry was performed using a Bruker Micro TOF ESI-TOF mass spectrometer. During the experiment, all pH values were measured by PH-10C pH meter. The purity of the synthesized derivatives was characterized by a purity test (as shown in Appendix A).

### 3.2. Synthesis

#### 3.2.1. Synthesis Method of Intermediate I

The indole (10.88 mmol) was added to the mixture of pyridine (14.2 mmol) and tetrahydrofuran (40 mL) at 0 °C, and the reaction was stirred at 0 °C for 0.5 h. Then, the mixed solution of trichloroacetyl chloride (14.2 mmol) and tetrahydrofuran (30 mL) was added slowly. After the reaction was raised to room temperature and stirred continuously for 16 h, 10% HCl solution was added to the reaction system at 0 °C, and the organic phase was extracted with ethyl acetate (300 mL), washed with salt water, and dried with anhydrous magnesium sulfate. The solvent was removed on the rotary evaporator, and the solid product was directly dissolved in anhydrous methanol (150 mL). Then, 10% KOH solution (20 mL) was added, the reaction mixture was heated and reflux for 5 h, and, then, concentrated in the rotary evaporator, the solvent was removed, and the large solid product was extracted with ethyl acetate (200 mL). The organic phase was washed with salt water, dried with anhydrous magnesium sulfate, and the solvent was removed. The crude product was purified by silica gel column chromatography with the eluent of n-hexane/ethyl acetate (4/1). Finally, Intermediate I was obtained.

##### Methyl 6-chloro-1H-indole-3-carboxylate (Intermediate I)

Yield, 85%; white solid, m.p. 182–185 °C (DMSO); ^1^H NMR (400 MHz, DMSO-*d*_6_) *δ* 12.05 (s, 1H), 8.14 (d, *J* = 3.0 Hz, 1H), 7.99 (d, *J* = 8.5 Hz, 1H), 7.55 (d, *J* = 2.0 Hz, 1H), 7.22 (d, *J* = 8.5 Hz, 1H), 3.82 (s, 3H). ^13^C NMR (100 MHz, DMSO-*d*_6_) *δ* 164.89, 137.24, 133.90, 127.51, 124.86, 122.20, 122.09, 112.51, 107.03, 51.25. MS-ESI *m*/*z*: calcd for C_10_H_8_ClN_2_O[M + H]+: 210.0244; found: 210.1683.

#### 3.2.2. Synthesis Method of Intermediate II

Under the protection of inert gas, intermediate I (2.5 mmol) was dissolved in dichloromethane (5 mL) and N-chlorosuccinimide (NCS, 2.75 mmol), and N, N-dimethylpiperazine (1.25 mmol) dichloromethane (1 mL) solution was added drop-by-drop at 0 °C. The reaction system was stirred continuously at 0 °C for 2 h. Then, a mixed solution of trichloroacetic acid (0.63 mmol) and N-methylaniline (5 mmol) dichloromethane (2 mL) was added. The reaction was then heated to room temperature and stirred continuously for 3.5 h at room temperature. At the end of the reaction, the reaction mixture was washed with saturated sodium bicarbonate aqueous solution, 1N concentration of hydrochloric acid, and salt water, and then, the separated organic phase was dried by anhydrous magnesium sulfate and condensed under pressure on a rotary evaporator. Finally, the solid product was purified by column chromatography with the eluent of n-hexane/ethyl acetate (7/1), and the intermediate II was finally obtained.

##### Methyl 6-chloro-2-(methyl(phenyl)amino)-1H-indole-3-carboxylate (Intermediate II)

Yield, 64%; white solid, m.p. 154–156 °C (DMSO); ^1^H NMR (400 MHz, DMSO-d_6_) δ 12.08 (s, 1H), 7.92 (d, J = 8.5 Hz, 1H), 7.34 (d, J = 1.9 Hz, 1H), 7.28–7.20 (m, 2H), 7.17 (d, J = 8.5 Hz, 1H), 6.86 (t, J = 7.3 Hz, 1H), 6.83–6.77 (m, 2H), 3.65 (s, 3H), 3.37 (s, 3H). ^13^C NMR (100 MHz, DMSO-d_6_) δ 163.77, 148.66, 147.76, 133.48, 129.46 (2C), 126.94, 125.51, 122.26, 121.79, 120.10, 115.75 (2C), 111.38, 97.10, 50.93, 40.15. MS-ESI *m*/*z*: calcd for C_17_H_15_ClN_2_O_2_[M + H]+: 315.0822; found: 315.1015.

#### 3.2.3. Synthesis Method of Intermediate III

An appropriate amount of diphenyl ether was added to the round bottom flask, and then, the reactor temperature was heated to about 60 °C, until the diphenyl ether was in the molten state, and then, intermediate II was added to the diphenyl ether. Then, the temperature increased to 250 °C reflux for 1–3 h, the end of the reaction, until the temperature was reduced to room temperature, and reactants were dumped into a large quantity of petroleum ether. The intermediate III was obtained by collecting precipitated precipitation.

#### 3.2.4. Synthesis Method of Intermediate IV

Under nitrogen protection, Dry intermediates III was added to the phosphorus oxychloride, reflux, under 110 °C for about 12 h of reaction. After the reaction is cooled to room temperature, phosphorus oxychloride was removed on the rotary evaporator. Then, ice water was added to the residue, the pH was adjusted to 9.0 with sodium bicarbonate, and the temperature during the period did not exceed 40 °C. The organic phase was extracted with dichloromethane, and the crude product was concentrated on a rotary evaporator. The crude product was first purified by silica gel column chromatography with the eluent of petroleum ether/ethyl acetate (2/1) to remove other impurities and, then, purified with the eluent of dichloromethane/methanol (40/1) to obtain orange–red product Intermediate **IV**.

##### 8,11-dichloro-5-methyl-5H-indolo[2,3-b]quinoline (Intermediate IV)

Yield, 64%; white solid, m.p. 227–229 °C (DMSO); ^1^H NMR (400 MHz, Chloroform-*d*) *δ* 8.29 (d, *J* = 8.2 Hz, 1H), 8.11 (d, *J* = 8.2 Hz, 1H), 7.75 (t, *J* = 7.8 Hz, 1H), 7.61 (d, *J* = 8.6 Hz, 1H), 7.52 (d, *J* = 1.8 Hz, 1H), 7.46 (t, *J* = 7.6 Hz, 1H), 7.07 (d, *J* = 8.2 Hz, 1H), 4.21 (s, 3H). ^13^C NMR (100 MHz, Chloroform-*d*) *δ* 155.82, 155.77, 136.71, 136.06, 135.13, 131.24, 125.94, 124.16, 123.75, 122.53, 122.01, 120.23, 119.06, 117.64, 114.31, 33.17. MS-ESI *m*/*z*: calcd for C_16_H_10_Cl_2_N_2_ [M + H]+: 301.0221; found: 301.1502.

#### 3.2.5. Synthesis Method of Intermediate Phenylhydrazine

The purchased phenylhydrazine hydrochloride was dissolved in dichloromethane, and 3 M potassium hydroxide was added to carry out the reaction at room temperature, with attention to the whole process to avoid light, and the reaction lasted for about 1–3 h. The organic phase was extracted with dichlorohexane, and the solvent was removed on a rotary evaporator at low pressure. The temperature of solvent removal was not more than 40 °C. When the solvent was removed by rotary evaporation, the white solid product was phenylhydrazine.

#### 3.2.6. Synthesis Method of Target Compounds **A1–A10**

The Intermediate IV was dissolved in N, N-dimethylformamide (DMF), and then, the corresponding alcohol hydroxyl compounds were added, and the reaction was reflux at 110 °C for 3 h. After the reaction was over, the organic phase was extracted with dichloromethane and, then, washed with salt water. After that, the organic phase was dried with anhydrous magnesium sulfate, and the solvent was removed at low pressure on the rotary evaporator. The crude product was first purified by silica gel column chromatography with the ratio of petroleum ether/ethyl acetate (2/1) eluent to remove other impurities, and then, it was purified with the ratio of dichloromethane/methanol (40/1). Finally, the target compound was obtained.

##### 8-chloro-5-methyl-11-phenoxy-5H-indolo[2,3-b]quinoline (**A1**)

Yield, 81%; brown solid, m.p. 195–196 °C (DMSO); ^1^H NMR (400 MHz, Chloroform-*d*) *δ* 8.23–8.18 (m, 1H), 7.81 (d, *J* = 6.3 Hz, 2H), 7.66 (s, 1H), 7.43 (d, *J* = 8.1 Hz, 1H), 7.32 (d, *J* = 2.4 Hz, 1H), 7.24–7.17 (m, 1H), 7.12–7.07 (m, 1H), 7.01 (d, *J* = 1.3 Hz, 1H), 7.00–6.94 (m, 2H), 6.93–6.89 (m, 1H), 6.89–6.83 (m, 1H), 4.36 (s, 3H). ^13^C NMR (100 MHz, Chloroform-*d*) *δ* 161.65, 157.40, 156.01, 153.26, 150.68, 137.10, 133.47, 130.50, 129.16, 128.51, 123.61, 123.44, 122.30, 121.43, 119.47, 119.40, 118.76, 116.40, 116.25, 114.56, 114.50, 113.53, 32.46. MS-ESI *m*/*z*: calcd for C_22_H_15_ClN_2_O[M + H]+: 359.0873; found: 359.1959.

##### 8-chloro-11-(2-fluorophenoxy)-5-methyl-5H-indolo[2,3-b]quinoline (**A2**)

Yield, 73%; orange solid, m.p. 325–328 °C (DMSO); ^1^H NMR (400 MHz, Chloroform-*d*) *δ* 8.23 (d, *J* = 8.2, 1.4 Hz, 1H), 7.95 (d, *J* = 8.7 Hz, 1H), 7.89 (d, *J* = 8.7 Hz, 1H), 7.60 (s, 1H), 7.51 (d, *J* = 8.1 Hz, 1H), 7.34 (d, *J* = 8.4 Hz, 1H), 7.31–7.25 (m, 1H), 7.11–7.04 (m, 1H), 7.00 (d, *J* = 8.3 Hz, 1H), 6.92–6.86 (m, 1H), 6.65 (t, *J* = 8.3 Hz, 1H), 4.37 (s, 3H). ^13^C NMR (100 MHz, Chloroform-*d*) *δ* 158.24, 156.33, 154.27, 152.54, 151.37, 148.13, 140.16, 133.84, 132.32, 131.82, 124.99, 124.30, 121.73, 119.94, 117.85, 117.11, 116.63, 116.14, 115.10, 114.68, 113.32, 34.09. MS-ESI *m*/*z*: calcd for C_22_H_14_ClFN_2_O[M + H]+: 377.0779; found: 377.1835.

##### 8-chloro-11-(3-fluorophenoxy)-5-methyl-5H-indolo[2,3-b]quinoline (**A3**)

Yield, 82%; orange solid, m.p. 250–252 °C (DMSO); ^1^H NMR (400 MHz, Chloroform-*d*) δ 8.18–8.10 (m, 1H), 8.01 (s, 1H), 7.88–7.77 (m, 2H), 7.66 (d, *J* = 1.8 Hz, 1H), 7.44 (d, *J* = 8.1 Hz, 1H), 7.39 (d, *J* = 8.2 Hz, 1H), 7.01 (d, *J* = 8.2 Hz, 1H), 6.85–6.80 (m, 1H), 6.80–6.68 (m, 2H), 4.37 (s, 3H). ^13^C NMR (100 MHz, Chloroform-*d*) *δ* 164.92, 162.59, 158.43, 154.66, 150.84, 138.14, 134.78, 131.62, 131.09 (d, *J* = 9.7 Hz), 130.21 (d, *J* = 10.2 Hz), 124.31 (d, *J* = 4.4 Hz), 122.55, 120.60, 120.27, 117.51, 117.00, 114.64, 111.15, 111.12, 110.33 (d, *J* = 21.2 Hz), 103.70 (d, *J* = 25.7 Hz), 33.45. MS-ESI *m*/*z*: calcd for C_22_H_14_ClFN_2_O[M + H]+: 377.0781; found: 377.1899.

##### 8-chloro-11-(4-fluorophenoxy)-5-methyl-5H-indolo[2,3-b]quinoline (**A4**)

Yield, 76%; yellow solid, m.p. 255–256 °C (DMSO); ^1^H NMR (400 MHz, Chloroform-*d*) *δ* 8.21–8.15 (m, 1H), 7.86–7.77 (m, 2H), 7.66 (s, 1H), 7.44 (d, *J* = 8.1 Hz, 1H), 7.32 (d, *J* = 8.3 Hz, 1H), 7.03–6.92 (m, 5H), 4.38 (s, 3H). ^13^C NMR (100 MHz, Chloroform-*d*) *δ* 158.63, 157.54, 156.22, 153.76, 152.03, 150.41, 137.15, 133.56, 130.49, 123.44, 123.33, 121.34, 119.38, 116.52, 116.13, 115.80, 115.72, 115.70, 115.62, 115.57, 115.52, 113.52, 32.34. MS-ESI *m*/*z*: calcd for C_22_H_14_ClFN_2_O[M + H]+: 377.0763; found: 377.1875.

##### 8-chloro-11-(2-methoxyphenoxy)-5-methyl-5H-indolo[2,3-b]quinoline (**A5**)

Yield, 78%; yellow solid, m.p. 244–247 °C (DMSO); ^1^H NMR (400 MHz, Chloroform-*d*) *δ* 8.25 (d, *J* = 8.2 Hz, 1H), 7.79 (d, *J* = 3.5 Hz, 2H), 7.67 (s, 1H), 7.41 (d, *J* = 8.1 Hz, 1H), 7.36 (d, *J* = 8.2 Hz, 1H), 7.13 (d, *J* = 8.2 Hz, 1H), 7.10–7.04 (m, 1H), 6.98 (d, *J* = 8.2 Hz, 1H), 6.69 (t, *J* = 7.8 Hz, 1H), 6.58 (d, *J* = 8.1 Hz, 1H), 4.39 (s, 3H), 4.05 (s, 3H). ^13^C NMR (100 MHz, Chloroform-*d*) *δ* 157.89, 154.14, 151.28, 148.00, 145.27, 137.12, 133.27, 130.27, 123.68, 123.37, 122.94, 121.14, 120.07, 119.84, 119.19, 116.46, 116.26, 115.49, 114.58, 113.34, 111.91, 55.27, 32.17. MS-ESI *m*/*z*: calcd for C_23_H_17_ClN_2_O_2_[M + H]+: 389.0979; found: 389.2214.

##### 8-chloro-11-(3-methoxyphenoxy)-5-methyl-5H-indolo[2,3-b]quinoline (**A6**)

Yield, 83%; orange solid, m.p. 223–224 °C (DMSO); ^1^H NMR (400 MHz, Chloroform-*d*) *δ* 8.21–8.16 (m, 1H), 7.80 (d, *J* = 6.2 Hz, 2H), 7.67 (s, 1H), 7.45–7.36 (m, 2H), 7.17 (t, *J* = 8.3 Hz, 1H), 7.00 (d, *J* = 8.2 Hz, 1H), 6.64 (d, *J* = 8.2 Hz, 1H), 6.60 (t, *J* = 2.4 Hz, 1H), 6.52 (d, *J* = 8.1 Hz, 1H), 4.37 (s, 3H), 3.74 (s, 4H). ^13^C NMR (100 MHz, Chloroform-*d*) *δ* 161.22, 158.58, 158.11, 154.72, 151.35, 138.14, 134.47, 131.41, 130.59, 124.59, 124.48, 122.32, 120.60, 120.41, 117.42, 117.33, 116.80, 114.47, 108.52, 107.70, 102.27, 55.45, 33.35. MS-ESI *m*/*z*: calcd for C_23_H_17_ClN_2_O_2_[M + H]+: 389.0961; found: 389.2096.

##### 8-chloro-11-(4-methoxyphenoxy)-5-methyl-5H-indolo[2,3-b]quinoline (**A7**)

Yield, 76%; yellow solid, m.p. 197–200 °C (DMSO); ^1^H NMR (400 MHz, Chloroform-*d*) *δ* 8.22 (d, *J* = 8.1 Hz, 1H), 7.80 (d, *J* = 7.5 Hz, 2H), 7.65 (s, 1H), 7.42 (d, *J* = 8.2 Hz, 1H), 7.30 (d, *J* = 8.2 Hz, 1H), 6.94 (d, *J* = 9.1 Hz, 2H), 6.86–6.79 (m, 3H), 6.75 (d, *J* = 8.8 Hz, 1H), 4.35 (s, 3H), 3.75 (s, 3H). ^13^C NMR (100 MHz, Chloroform-*d*) *δ* 158.64, 155.43, 154.49, 152.22, 151.18, 138.15, 134.31, 131.43, 124.71, 124.60, 122.31, 120.59, 120.36, 117.49, 117.30, 116.52 (2C), 116.16, 115.12 (2C), 114.78, 114.49, 58.36, 33.38. MS-ESI *m*/*z*: calcd for C_23_H_17_ClN_2_O_2_[M + H]+: 389.0975; found: 389.2216.

##### 8-chloro-11-(3,4-dimethoxyphenoxy)-5-methyl-5H-indolo[2,3-b]quinoline (**A8**)

Yield, 63%; dark brown solid, m.p. 245–249 °C (DMSO); ^1^H NMR (400 MHz, Chloroform-*d*) *δ* 8.23 (d, *J* = 8.3 Hz, 1H), 7.86–7.75 (m, 2H), 7.68 (s, 1H), 7.44 (d, *J* = 8.1 Hz, 1H), 7.32 (d, *J* = 8.2 Hz, 1H), 7.00 (d, *J* = 8.2 Hz, 1H), 6.80 (d, *J* = 2.9 Hz, 1H), 6.67 (d, *J* = 9.1 Hz, 1H), 6.31 (d, *J* = 8.8 Hz, 1H), 4.37 (s, 3H), 3.83–3.76 (m, 6H). ^13^C NMR (100 MHz, Chloroform-*d*) *δ* 158.51, 154.32, 151.96, 151.31, 150.30, 145.13, 138.15, 134.42, 131.49, 124.70, 124.64, 122.37, 120.57, 120.49, 117.45, 117.32, 114.52, 112.45, 111.72, 105.91, 105.89, 101.05, 56.30, 56.16, 33.47. MS-ESI *m*/*z*: calcd for C_24_H_19_ClN_2_O_3_[M + H]+: 419.1084; found: 419.2344.

##### 8-chloro-5-methyl-11-(p-tolyloxy)-5H-indolo[2,3-b]quinoline (**A9**)

Yield, 74%; brown solid, m.p. 219–220 °C (DMSO); ^1^H NMR (400 MHz, Chloroform-*d*) *δ* 8.23–8.18 (m, 1H), 7.82–7.78 (m, 2H), 7.66 (s, 1H), 7.41 (d, *J* = 8.1 Hz, 1H), 7.33 (d, *J* = 8.3 Hz, 1H), 7.09 (d, *J* = 8.5 Hz, 2H), 6.97 (d, *J* = 8.2 Hz, 1H), 6.92–6.85 (m, 2H), 4.36 (s, 3H), 2.30 (s, 3H). ^13^C NMR (100 MHz, Chloroform-*d*) *δ* 158.56, 155.07, 154.44, 151.97, 138.14, 134.33, 132.76, 131.44, 130.56 (2C), 129.91, 124.69, 124.54, 122.34, 120.56, 120.37, 117.49, 117.26, 116.44, 115.36 (2C), 114.51, 58.27, 18.36. MS-ESI *m*/*z*: calcd for C_23_H_17_ClN_2_O[M + H]+: 373.1029; found: 373.2255.

##### 8-chloro-11-(4-chlorophenoxy)-5-methyl-5H-indolo[2,3-b]quinoline (**A10**)

Yield, 85%; yellow solid, m.p. 258–262 °C (DMSO); ^1^H NMR (400 MHz, Chloroform-*d*) *δ* 8.19–8.12 (m, 1H), 7.83 (d, *J* = 6.2 Hz, 2H), 7.67 (s, 1H), 7.45 (d, *J* = 8.1 Hz, 1H), 7.37 (d, *J* = 8.1 Hz, 1H), 7.28 (s, 1H), 7.02 (d, *J* = 8.2 Hz, 1H), 6.97–6.92 (m, 2H), 6.89–6.81 (m, 1H), 4.38 (s, 3H). ^13^C NMR (100 MHz, Chloroform-*d*) *δ* 158.40, 155.54, 154.57, 151.09, 134.79, 131.63, 130.16, 129.40, 128.38, 124.39, 124.33, 122.54, 120.63, 120.26, 117.52, 117.07, 116.91, 116.78, 114.64, 33.48. MS-ESI *m*/*z*: calcd for C_22_H_14_Cl_2_N_2_O[M + H]+: 393.0483; found: 393.1729.

#### 3.2.7. Synthesis Method of Target Compounds **B1–B9**

The Intermediate IV was dissolved in N, N-dimethylformamide (DMF). Then, the corresponding phenylhydrazine compounds were added and reflux at 110 °C for 3–6 h. After the reaction, dichloromethane, ethyl acetate and salt water were added to the reaction system for washing, and solids were collected to obtain the target compounds **B1–B9**.

##### 8-chloro-5-methyl-11-(2-phenylhydrazineyl)-5H-indolo[2,3-b]quinoline (**B1**)

Yield, 70%; gray solid, m.p. > 400 °C (DMSO); ^1^H NMR (400 MHz, DMSO-*d*_6_) *δ* 13.86 (s, 1H), 10.96 (s, 1H), 9.16 (s, 1H), 8.82 (d, *J* = 8.6 Hz, 1H), 8.50 (d, *J* = 8.9 Hz, 1H), 8.16 (d, *J* = 8.7 Hz, 1H), 8.04 (t, *J* = 7.9 Hz, 1H), 7.91–7.74 (m, 1H), 7.69 (t, *J* = 7.8 Hz, 1H), 7.62 (s, 1H), 7.21 (m, 2H), 6.86 (m, 2H), 4.57 (s, 1H), 4.26 (s, 3H). ^13^C NMR (100 MHz, DMSO-*d*_6_) *δ* 152.82, 149.05, 148.47, 138.82, 137.00, 133.67, 130.70, 130.58, 129.77(2C), 124.76, 124.34, 122.34, 120.90, 120.28, 117.30, 113.87, 113.60(2C), 111.69, 97.44, 36.14. MS-ESI *m*/*z*: calcd for C_22_H_17_ClN_4_[M + H]^+^: 373.1142; found: 373.2594.

##### 8-chloro-11-(2-(2-fluorophenyl)hydrazineyl)-5-methyl-5H-indolo[2,3-b]quinoline (**B2**)

Yield, 56%; white solid, m.p. > 400 °C (DMSO); ^1^H NMR (400 MHz, Methanol-*d*_4_) *δ* 8.87–8.71 (m, 1H), 8.51 (d, *J* = 8.9 Hz, 1H), 8.20 (d, *J* = 8.8 Hz, 1H), 8.07 (t, *J* = 8.0 Hz, 1H), 7.72 (t, *J* = 7.8 Hz, 1H), 7.59 (s, 1H), 7.29–7.18 (m, 2H), 7.00 (t, *J* = 7.7 Hz, 1H), 6.93–6.82 (m, 2H), 4.24 (s, 3H). ^13^C NMR (100 MHz, Methanol-*d*_4_) *δ* 153.54, 149.86, 139.43, 137.74, 136.47, 134.43, 131.86, 126.02, 125.55, 123.16, 121.01, 117.82, 116.68, 116.50, 116.18, 114.74, 112.40, 98.61, 57.28, 36.21. MS-ESI *m*/*z*: calcd for C_22_H_16_ClFN_4_[M + H]^+^: 391.1048; found: 391.2159.

##### 8-chloro-11-(2-(3-fluorophenyl)hydrazineyl)-5-methyl-5H-indolo[2,3-b]quinoline (**B3**)

Yield, 73%; gray solid, m.p. > 400 °C (DMSO); ^1^H NMR (400 MHz, DMSO-*d*_6_) *δ* 8.77 (s, 1H), 8.47 (d, *J* = 8.8 Hz, 1H), 8.19 (d, *J* = 8.8 Hz, 1H), 8.06 (d, *J* = 8.5 Hz, 1H), 7.71 (t, *J* = 7.8 Hz, 1H), 7.57 (s, 1H), 7.32–7.19 (m, 2H), 6.76 (d, *J* = 8.1 Hz, 1H), 6.66 (m, 2H), 4.22 (s, 3H). ^13^C NMR (100 MHz, DMSO-*d*_6_) *δ* 152.56, 138.53, 136.91, 133.54, 131.32, 131.23, 131.09, 124.68, 122.46, 120.00, 116.89, 113.85, 111.50, 109.38, 107.29, 107.07, 100.53, 100.27, 97.70, 35.27. MS-ESI *m*/*z*: calcd for C_22_H_16_ClFN_4_[M + H]^+^: 391.1045; found: 391.2103.

##### 8-chloro-11-(2-(4-fluorophenyl)hydrazineyl)-5-methyl-5H-indolo[2,3-b]quinoline (**B4**)

Yield, 71%; yellow solid, m.p. > 400 °C (DMSO); ^1^H NMR (400 MHz, DMSO-*d*_6_) *δ* 8.73 (s, 1H), 8.55 (d, *J* = 8.8 Hz, 1H), 8.19 (d, *J* = 8.8 Hz, 1H), 8.06 (t, *J* = 7.9 Hz, 1H), 7.70 (t, *J* = 7.8 Hz, 1H), 7.58 (s, 1H), 7.27–7.20 (m, 1H), 7.05 (t, *J* = 8.7 Hz, 2H), 6.94 (d, *J* = 9.0 Hz, 2H), 4.25 (s, 3H). ^13^C NMR (100 MHz, DMSO) *δ* 153.54, 144.97, 138.62, 135.38, 134.13, 131.84, 129.43, 127.46, 125.78, 125.61, 122.51, 121.45, 117.37(2C), 116.74(2C), 116.51, 115.73, 115.64, 114.31, 111.99, 33.53. MS-ESI *m*/*z*: calcd for C_22_H_16_ClFN_4_[M + H]^+^: 391.1061; found: 391.2028.

##### 8-chloro-11-(2-(2,4-difluorophenyl)hydrazineyl)-5-methyl-5H-indolo[2,3-b]quinoline (**B5**)

Yield, 56%; white solid, m.p. > 400 °C (DMSO); ^1^H NMR (400 MHz, Methanol-*d*_4_) *δ* 8.76 (s, 1H), 8.50 (d, *J* = 8.8 Hz, 1H), 8.20 (d, *J* = 8.7 Hz, 1H), 8.12–8.01 (m, 1H), 7.72 (t, *J* = 7.7 Hz, 1H), 7.60 (s, 1H), 7.24 (m, 2H), 6.90 (m, 2H), 4.25 (s, 3H). ^13^C NMR (100 MHz, CD_3_OD_SPE) *δ* 153.30, 152.37, 150.85, 149.28, 141.92, 139.78, 138.11, 134.37, 131.96, 128.69, 127.04, 125.50, 123.72, 120.55, 118.96, 116.90, 115.40, 114.75, 113.19, 107.73, 104.57, 36.00. MS-ESI *m*/*z*: calcd for C_22_H_15_ClF_2_N_4_[M + H]^+^: 409.0953; found: 409.2040.

##### 8-chloro-5-methyl-11-(2-(o-tolyl)hydrazineyl)-5H-indolo[2,3-b]quinoline (**B6**)

Yield, 74%; white solid, m.p. 255–256 °C (DMSO); ^1^H NMR (400 MHz, DMSO-*d*_6_) *δ* 8.82 (s, 1H), 8.46 (d, *J* = 9.3 Hz, 1H), 8.20 (d, *J* = 8.7 Hz, 1H), 8.07 (t, *J* = 8.0 Hz, 1H), 7.72 (t, *J* = 7.7 Hz, 1H), 7.59 (s, 1H), 7.22 (d, *J* = 16.6 Hz, 2H), 7.04 (t, *J* = 7.7 Hz, 1H), 6.83 (t, *J* = 7.4 Hz, 1H), 6.74 (d, *J* = 7.9 Hz, 1H), 4.24 (s, 3H), 3.51 (s, 3H). ^13^C NMR (100 MHz, DMSO) *δ* 153.17, 148.78, 145.56, 138.03, 133.49, 133.00, 131.11, 130.93, 129.97, 127.17, 124.35, 123.54, 122.84, 122.11, 120.95, 119.56, 117.02, 114.44, 113.76, 112.53, 111.60, 33.86, 20.16. MS-ESI *m*/*z*: calcd for C_23_H_19_ClN_4_[M + H]^+^: 387.1298; found: 387.2787.

##### 8-chloro-5-methyl-11-(2-(m-tolyl)hydrazineyl)-5H-indolo[2,3-b]quinoline (**B7**)

Yield, 80%; gray solid, m.p. > 400 °C (DMSO); ^1^H NMR (400 MHz, DMSO-*d*_6_) *δ* 8.78 (s, 1H), 8.55 (d, *J* = 8.8 Hz, 1H), 8.18 (d, *J* = 8.8 Hz, 1H), 8.09–8.02 (m, 1H), 7.70 (t, *J* = 7.8 Hz, 1H), 7.57 (s, 1H), 7.22 (d, *J* = 9.0 Hz, 1H), 7.13 (t, *J* = 7.8 Hz, 1H), 6.76 (d, *J* = 2.3 Hz, 1H), 6.72 (d, *J* = 7.5 Hz, 2H), 4.23 (s, 3H), 2.23 (s, 3H). ^13^C NMR (100 MHz, DMSO-*d*_6_) *δ* 152.79, 148.12, 139.19, 138.44, 136.94, 133.49, 130.94, 129.46, 124.61, 122.38, 121.90, 120.24, 116.85, 114.06, 113.85, 111.40, 110.85, 97.50, 56.51, 35.17, 18.18. MS-ESI *m*/*z*: calcd for C_23_H_19_ClN_4_[M + H]^+^: 387.1291; found: 387.2803.

##### 8-chloro-5-methyl-11-(2-(p-tolyl)hydrazineyl)-5H-indolo[2,3-b]quinoline (**B8**)

Yield, 71%; light yellow solid, m.p. > 400 °C (DMSO); ^1^H NMR (400 MHz, DMSO-*d*_6_) *δ* 8.76 (s, 1H), 8.57 (d, *J* = 8.8 Hz, 1H), 8.17 (d, *J* = 8.7 Hz, 1H), 8.05 (d, *J* = 8.5 Hz, 1H), 7.70 (m, 1H), 7.57 (s, 1H), 7.21 (d, *J* = 8.9 Hz, 1H), 7.06 (d, *J* = 8.1 Hz, 2H), 6.88–6.78 (m, 2H), 4.23 (s, 3H), 2.21 (s, 3H). ^13^C NMR (100 MHz, DMSO) *δ* 152.81, 148.40, 145.69, 138.41, 136.92, 134.11, 130.93, 130.85, 129.94(2C), 128.38, 124.58, 122.35, 120.18, 116.79, 113.71(2C), 111.35, 100.65, 98.74, 97.10, 33.57, 19.93. MS-ESI *m*/*z*: calcd for C_23_H_19_ClN_4_[M + H]^+^: 387.1285; found: 387.2360.

##### 8-chloro-11-(2-(4-methoxyphenyl)hydrazineyl)-5-methyl-5H-indolo[2,3-b]quinoline (**B9**)

Yield, 70%; brownish red solid, m.p. > 400 °C (DMSO); ^1^H NMR (400 MHz, DMSO-*d*_6_) *δ* 8.77 (s, 1H), 8.59 (d, *J* = 8.8 Hz, 1H), 8.21 (d, *J* = 8.8 Hz, 1H), 8.16 (d, *J* = 8.9 Hz, 1H), 8.04 (t, *J* = 8.0 Hz, 1H), 7.68 (t, *J* = 7.7 Hz, 1H), 7.56 (s, 1H), 7.31 (d, *J* = 8.7 Hz, 0H), 7.26–7.17 (m, 1H), 6.93–6.80 (m, 4H), 4.42 (s, 1H), 4.22 (s, 3H), 3.68 (s, 3H). ^13^C NMR (100 MHz, DMSO) *δ* 154.59, 151.36, 148.39, 147.17, 141.69, 139.42, 138.20, 136.66, 133.47, 130.83, 127.22, 124.56, 122.82, 120.59, 116.85, 115.21(2C), 114.96(2C), 113.84, 111.08, 54.29, 34.56. MS-ESI *m*/*z*: calcd for C_23_H_19_ClN_4_O[M + H]^+^: 403.1247; found: 403.2404.

#### 3.2.8. Synthesis Method of the Target Compounds **C1–C10**, **D1–D6**

The Intermediate IV was dissolved in N, N-dimethylformamide (DMF). Then, the corresponding amide compound was added, and the reaction was reflux at 110 °C for 3–6 h. After the reaction, dichloromethane, ethyl acetate, and salt water were added to the reaction system to wash, and solids were collected to obtain the target compound.

##### N-(8-chloro-5-methyl-5H-indolo[2,3-b]quinolin-11-yl)acetamide (**C1**)

Yield, 83%; yellow solid, m.p. 324–325 °C (DMSO); ^1^H NMR (400 MHz, DMSO-*d*_6_) *δ* 8.25 (d, *J* = 8.3 Hz, 1H), 7.98 (d, *J* = 8.7 Hz, 1H), 7.86 (t, *J* = 7.6 Hz, 1H), 7.75 (d, *J* = 8.2 Hz, 1H), 7.55 (s, 1H), 7.53 (s, 1H), 7.14 (d, *J* = 8.1 Hz, 1H), 4.29 (s, 3H), 2.32 (s, 3H). ^13^C NMR (100 MHz, DMSO-*d*_6_) *δ* 169.34, 157.58, 155.96, 137.63, 132.79, 131.55, 126.10, 125.27, 122.69, 122.23, 120.05, 119.93, 119.20, 118.92, 116.85, 115.74, 33.40, 23.88. MS-ESI *m*/*z*: calcd for C_18_H_14_ClN_3_O[M + H]^+^: 324.0825; found: 324.1824.

##### N-(8-chloro-5-methyl-5H-indolo[2,3-b]quinolin-11-yl)benzamide (**C2**)

Yield, 89%; yellow solid, m.p. 275–277 °C (DMSO); ^1^H NMR (400 MHz, DMSO-*d*_6_) *δ* 11.96 (s, 1H), 8.57 (d, *J* = 8.5 Hz, 1H), 8.46 (d, *J* = 8.8 Hz, 1H), 8.30 (d, *J* = 7.6 Hz, 2H), 8.20 (t, *J* = 8.0 Hz, 1H), 7.86 (s, 1H), 7.81–7.74 (m, 4H), 7.74–7.64 (m, 2H), 7.51 (d, *J* = 8.5 Hz, 1H), 4.58 (s, 3H). ^13^C NMR (100 MHz, DMSO-*d*_6_) *δ* 166.35, 149.50, 143.45, 141.47, 137.27, 134.48, 134.27, 133.44, 133.12, 129.36 (2C), 129.03 (2C), 126.96, 126.22, 126.08, 123.81, 121.10, 118.86, 117.74, 116.96, 113.04, 37.82. MS-ESI *m*/*z*: calcd for C_23_H_16_ClN_3_O[M + H]^+^: 386.0982; found: 387.2003.

##### N-(8-chloro-5-methyl-5H-indolo[2,3-b]quinolin-11-yl)-2-fluorobenzamide (**C3**)

Yield, 64%; brownish red solid, m.p. 361–362 °C (DMSO); ^1^H NMR (400 MHz, DMSO-*d*_6_) *δ* 11.94 (s, 1H), 8.54 (d, *J* = 8.4 Hz, 1H), 8.40 (d, *J* = 8.8 Hz, 1H), 8.15 (m, 1H), 8.01 (t, *J* = 7.4, 1H), 7.90 (d, *J* = 8.4 Hz, 1H), 7.85 (t, *J* = 7.7 Hz, 1H), 7.81 (s, 1H), 7.79–7.72 (m, 1H), 7.51 (m, 3H), 4.55 (s, 3H). ^13^C NMR (100 MHz, DMSO-*d*_6_) *δ* 163.70, 161.20, 158.71, 141.87, 137.34, 134.39, 134.33, 134.25, 133.87, 130.97, 126.41, 126.14, 125.72, 125.46 (d, *J* = 3.3 Hz), 123.07, 121.71, 120.41, 119.44, 117.52, 117.17, 116.95, 113.78, 37.14. MS-ESI *m*/*z*: calcd for C_23_H_15_ClFN_3_O[M + H]^+^: 404.0888; found: 404.1265.

##### N-(8-chloro-5-methyl-5H-indolo[2,3-b]quinolin-11-yl)-3-fluorobenzamide (**C4**)

Yield, 77%; light yellow solid, m.p. > 400 °C; ^1^H NMR (400 MHz, DMSO-*d*_6_) *δ* 11.98 (s, 1H), 8.66–8.49 (m, 1H), 8.43 (d, *J* = 8.9 Hz, 1H), 8.28–8.00 (m, 3H), 7.92–7.68 (m, 5H), 7.56 (d, *J* = 62.1 Hz, 1H), 4.55 (s, 3H). ^13^C NMR (100 MHz, DMSO-*d*_6_) *δ* 165.07, 159.83, 155.64, 143.30, 137.30, 134.49, 134.85, 134.50, 131.56, 129.78, 126.86, 126.07, 125.89, 125.19 (d, *J* = 5.4 Hz), 123.50, 122.07, 120.84, 118.93, 117.57, 115.95, 115.71, 113.50, 37.26. MS-ESI *m*/*z*: calcd for C_23_H_15_ClFN_3_O[M + H]^+^: 404.0876; found: 404.1280.

##### N-(8-chloro-5-methyl-5H-indolo[2,3-b]quinolin-11-yl)-4-fluorobenzamide (**C5**)

Yield, 71%; yellow solid, m.p. > 400 °C (DMSO); ^1^H NMR (400 MHz, DMSO-*d*_6_) *δ* 11.98 (s, 1H), 8.57 (d, *J* = 8.4 Hz, 1H), 8.46 (d, *J* = 8.9 Hz, 1H), 8.39 (d, *J* = 8.6 Hz, 2H), 8.21 (t, *J* = 8.0 Hz, 1H), 7.88 (t, *J* = 7.9 Hz, 1H), 7.84 (s, 1H), 7.73 (d, *J* = 8.5 Hz, 1H), 7.59–7.45 (m, 3H), 4.57 (s, 3H). ^13^C NMR (100 MHz, DMSO-*d*_6_) *δ* 165.28, 159.56, 149.61, 143.39, 143.26, 137.27, 134.53, 134.30, 131.97 (2C, d, *J* = 9.7 Hz), 129.63, 126.97, 126.18 (d, *J* = 9.1 Hz), 123.82, 121.05, 118.85, 117.74, 116.98, 116.49 (2C), 116.27, 113.06, 37.73. MS-ESI *m*/*z*: calcd for C_23_H_15_ClFN_3_O[M + H]^+^: 404.0891; found: 404.1273.

##### N-(8-chloro-5-methyl-5H-indolo[2,3-b]quinolin-11-yl)-2-methoxybenzamide (**C6**)

Yield, 65%; orange solid, m.p. 250–252 °C (DMSO); ^1^H NMR (400 MHz, DMSO-*d*_6_) *δ* 11.51 (s, 1H), 8.52 (d, *J* = 8.3 Hz, 1H), 8.38 (d, *J* = 8.8 Hz, 1H), 8.15 (t, *J* = 7.9 Hz, 1H), 8.00 (d, *J* = 8.4 Hz, 1H), 7.87 (t, *J* = 7.0 Hz, 2H), 7.80 (s, 1H), 7.66 (t, *J* = 7.9 Hz, 1H), 7.53 (d, *J* = 8.4 Hz, 1H), 7.36 (d, *J* = 8.4 Hz, 1H), 7.20 (t, *J* = 7.5 Hz, 1H), 4.54 (s, 3H), 4.08 (s, 3H). ^13^C NMR (100 MHz, DMSO-*d*_6_) *δ* 165.38, 157.34, 141.73, 137.27, 134.21, 133.74, 133.53, 130.50, 129.15, 126.55, 126.44, 125.66, 124.01, 123.05, 121.30, 120.61, 119.69, 117.45, 113.64, 112.73, 104.16, 99.82, 56.61, 37.11. MS-ESI *m*/*z*: calcd for C_24_H_18_ClN_3_O_2_[M + H]^+^: 416.1088; found: 416.1328.

##### N-(8-chloro-5-methyl-5H-indolo[2,3-b]quinolin-11-yl)-3-methoxybenzamide (**C7**)

Yield, 69%; yellow solid, m.p. 240–241 °C (DMSO); ^1^H NMR (400 MHz, DMSO-*d*_6_) *δ* 11.95 (s, 1H), 8.59–8.51 (m, 1H), 8.44 (d, *J* = 8.8 Hz, 1H), 8.19 (t, *J* = 8.7, 7.1, 1.5 Hz, 1H), 7.90–7.84 (m, 3H), 7.82 (s, 1H), 7.74 (d, *J* = 8.4 Hz, 1H), 7.61 (t, *J* = 8.0 Hz, 1H), 7.51 (d, *J* = 8.4 Hz, 1H), 7.34 (d, *J* = 8.3 Hz, 1H), 4.57 (s, 3H), 3.92 (s, 4H). ^13^C NMR (100 MHz, DMSO-*d*_6_) δ 166.06, 159.98, 140.92, 137.28, 134.45, 134.20, 133.12, 130.56, 128.73, 126.94, 126.07, 125.39, 124.34, 123.70, 121.23, 121.02, 119.37, 117.68, 113.97, 113.20, 105.64, 99.99, 56.04, 37.67. MS-ESI *m*/*z*: calcd for C_24_H_18_ClN_3_O_2_[M + H]^+^: 416.1081; found: 416.1336.

##### N-(8-chloro-5-methyl-5H-indolo[2,3-b]quinolin-11-yl)-4-methoxybenzamide (**C8**)

Yield, 52%; brown solid, m.p. 310–312 °C (DMSO); ^1^H NMR (400 MHz, DMSO-*d*_6_) *δ* 11.76 (s, 1H), 8.55 (d, *J* = 8.5 Hz, 1H), 8.45 (d, *J* = 8.8 Hz, 1H), 8.28 (d, *J* = 8.4 Hz, 2H), 7.99–7.84 (m, 1H), 7.83 (s, 1H), 7.69 (d, *J* = 8.5 Hz, 1H), 7.49 (d, *J* = 9.3 Hz, 1H), 7.22 (d, *J* = 8.4 Hz, 3H), 4.56 (s, 3H), 3.92 (s, 3H). ^13^C NMR (100 MHz, DMSO-*d*_6_) *δ* 164.13, 159.40, 141.21, 137.57, 134.20, 134.09, 133.57, 131.17 (2C), 128.24, 126.53, 126.38, 124.10, 122.41, 121.53, 119.93, 117.15, 114.62 (2C), 113.74, 105.36, 99.87, 56.15, 37.66. MS-ESI *m*/*z*: calcd for C_24_H_18_ClN_3_O_2_[M + H]^+^: 416.1063; found: 416.1347.

##### N-(8-chloro-5-methyl-5H-indolo[2,3-b]quinolin-11-yl)hexanamide (**C9**)

Yield, 73%; white solid, m.p. 284–286 °C (DMSO); ^1^H NMR (400 MHz, DMSO-*d*_6_) *δ* 11.64 (s, 1H), 8.54 (d, *J* = 8.4 Hz, 1H), 8.40 (d, *J* = 9.0 Hz, 1H), 8.16 (t, *J* = 8.1 Hz, 1H), 7.92–7.70 (m, 3H), 7.51 (d, *J* = 8.6 Hz, 1H), 4.53 (s, 3H), 2.83 (t, *J* = 8.0 Hz, 2H), 1.77 (m, 2H), 1.40 (m, 4H), 0.94 (t, *J* = 7.0 Hz, 3H). ^13^C NMR (100 MHz, DMSO-*d*_6_) *δ* 172.28, 155.32, 141.07, 137.62, 135.13, 130.64, 129.75, 127.92, 126.73, 126.48, 125.11, 123.61, 120.74, 117.65, 113.30, 102.48, 36.16, 31.52, 24.93, 22.40, 14.42. MS-ESI *m*/*z*: calcd for C_22_H_22_ClN_3_O[M + H]^+^: 380.1451; found: 380.1520.

##### N-(8-chloro-5-methyl-5H-indolo[2,3-b]quinolin-11-yl)-3-methylbutanamide (**C10**)

Yield, 81%; white solid, m.p. > 400 °C (DMSO); ^1^H NMR (400 MHz, DMSO-*d*_6_) *δ* 11.56 (s, 1H), 8.49 (d, *J* = 8.3 Hz, 1H), 8.41 (d, *J* = 8.9 Hz, 1H), 8.17 (t, *J* = 8.0 Hz, 1H), 7.88 (m, 2H), 7.80 (s, 1H), 7.52 (d, *J* = 8.4 Hz, 1H), 4.51 (s, 3H), 2.82–2.61 (m, 2H), 2.33–2.17 (m, 1H), 1.08 (d, *J* = 6.7 Hz, 6H). ^13^C NMR (100 MHz, DMSO-*d*_6_) *δ* 171.64, 156.39, 142.78, 137.24, 134.07, 131.54, 129.15, 128.32, 126.71, 126.48, 124.85, 123.35, 120.63, 119.24, 117.69, 100.58, 45.12, 34.61, 25.83, 22.93 (2C). MS-ESI *m*/*z*: calcd for C_21_H_20_ClN_3_O[M + H]^+^: 366.1295; found: 366.1364.

##### N-(8-chloro-5-methyl-5H-indolo[2,3-b]quinolin-11-yl)-2-fluorobenzenesulfonamide (**D1**)

Yield, 73%; yellow solid, m.p. 302–305 °C (DMSO); ^1^H NMR (400 MHz, DMSO-*d*_6_) *δ* 12.82 (s, 1H), 8.85 (d, *J* = 8.3 Hz, 1H), 8.12 (d, *J* = 8.3 Hz, 1H), 8.01 (d, *J* = 8.8 Hz, 1H), 7.96 (t, *J* = 5.2 Hz, 1H), 7.89 (t, *J* = 7.6 Hz, 1H), 7.59 (t, *J* = 7.3 Hz, 1H), 7.56–7.50 (m, 1H), 7.50 (s, 1H), 7.32 (m, 2H), 7.13 (d, *J* = 8.4 Hz, 1H), 4.16 (s, 3H). ^13^C NMR (100 MHz, DMSO-*d*_6_) *δ* 162.75, 159.84, 157.34, 155.04, 148.19, 138.47, 138.30, 133.46 (d, *J* = 8.0 Hz), 132.61, 129.91, 129.63, 128.09, 124.54 (d, *J* = 3.5 Hz), 123.88, 123.20, 121.82 (d, *J* = 5.0 Hz), 117.30, 117.09, 116.32, 111.58, 107.23, 35.20. MS-ESI *m*/*z*: calcd for C_22_H_15_ClFN_3_O_2_S[M + H]^+^: 440.0547; found: 440.1713.

##### N-(8-chloro-5-methyl-5H-indolo[2,3-b]quinolin-11-yl)-3-fluorobenzenesulfonamide (**D2**)

Yield, 69%; white solid, m.p. 321–323 °C (DMSO); ^1^H NMR (400 MHz, DMSO-*d*_6_) *δ* 12.73 (s, 1H), 8.87 (d, *J* = 8.4 Hz, 1H), 8.04 (d, *J* = 8.5 Hz, 1H), 7.99 (d, *J* = 8.6 Hz, 1H), 7.88 (t, *J* = 7.9 Hz, 1H), 7.72 (d, *J* = 7.7 Hz, 1H), 7.63–7.56 (m, 2H), 7.53 (t, *J* = 7.6 Hz, 1H), 7.48 (s, 1H), 7.40 (t, *J* = 8.6, 2.7 Hz, 1H), 7.13 (d, *J* = 8.4 Hz, 1H), 4.15 (s, 3H). ^13^C NMR (100 MHz, DMSO-*d*_6_) *δ* 163.31, 160.86, 154.96, 149.91 (d, *J* = 6.0 Hz), 147.97, 138.32, 132.65, 131.36 (d, *J* = 7.9 Hz), 129.95, 129.66, 124.02, 123.27, 122.12, 121.88 (d, *J* = 2.6 Hz), 121.56, 117.84 (d, *J* = 20.9 Hz), 116.39, 112.69, 112.46, 111.51, 107.10, 35.25. MS-ESI *m*/*z*: calcd for C_22_H_15_ClFN_3_O_2_S[M + H]^+^: 440.0558; found: 440.1708.

##### N-(8-chloro-5-methyl-5H-indolo[2,3-b]quinolin-11-yl)-4-fluorobenzenesulfonamide (**D3**)

Yield, 75%; white solid, m.p. > 400 °C (DMSO); ^1^H NMR (400 MHz, DMSO-*d*_6_) *δ* 12.71 (s, 1H), 8.96 (d, *J* = 8.4 Hz, 1H), 8.04 (d, *J* = 8.4 Hz, 1H), 7.99 (d, *J* = 8.7 Hz, 1H), 7.93 (m, 2H), 7.87 (d, *J* = 8.3 Hz, 1H), 7.53 (t, *J* = 7.7 Hz, 1H), 7.48 (s, 1H), 7.36 (t, *J* = 8.8 Hz, 2H), 7.15 (d, *J* = 8.4 Hz, 1H), 4.14 (s, 3H). ^13^C NMR (100 MHz, DMSO-*d*_6_) *δ* 164.65, 162.19, 153.50, 149.74, 147.83, 144.10, 138.42, 137.97, 132.63, 129.84 (d, *J* = 7.4 Hz), 128.40 (d, *J* = 9.0 Hz, 2C), 124.02, 123.18, 121.96, 121.37, 116.38, 116.06 (2C), 115.84, 111.45, 35.20. MS-ESI *m*/*z*: calcd for C_22_H_15_ClFN_3_O_2_S[M + H]^+^: 440.0543; found: 440.1703.

##### N-(8-chloro-5-methyl-5H-indolo[2,3-b]quinolin-11-yl)-2-methylbenzenesulfonamide (**D4**)

Yield, 67%; yellow solid, m.p. > 400 °C (DMSO); ^1^H NMR (400 MHz, DMSO-*d*_6_) *δ* 12.64 (s, 1H), 9.05 (d, *J* = 8.4 Hz, 1H), 7.98 (m, 3H), 7.88 (t, *J* = 7.9 Hz, 1H), 7.54 (t, *J* = 7.6 Hz, 1H), 7.49 (s, 1H), 7.46 (d, *J* = 7.6 Hz, 1H), 7.36 (d, *J* = 13.3 Hz, 2H), 7.07 (d, *J* = 8.4 Hz, 1H), 4.14 (s, 3H). ^13^C NMR (100 MHz, DMSO-*d*_6_) *δ* 154.98, 147.81, 145.37, 138.55, 136.23, 132.58, 132.25, 131.23, 130.11, 129.55, 126.33, 126.10, 124.72, 123.57, 123.00, 122.47, 121.73, 121.33, 116.34, 111.50, 106.04, 35.09, 20.48. MS-ESI *m*/*z*: calcd for C_23_H_18_ClN_3_O_2_S[M + H]^+^: 436.0808; found: 436.1984.

##### N-(8-chloro-5-methyl-5H-indolo[2,3-b]quinolin-11-yl)-4-methylbenzenesulfonamide (**D5**)

Yield, 84%; white solid, m.p. > 400 °C (DMSO); ^1^H NMR (400 MHz, DMSO-*d*_6_) *δ* 12.67 (s, 1H), 9.02 (d, J = 8.2 Hz, 1H), 8.10 (d, *J* = 8.3 Hz, 1H), 7.97 (d, *J* = 8.6 Hz, 1H), 7.87 (t, *J* = 7.8 Hz, 1H), 7.79 (d, *J* = 7.8 Hz, 2H), 7.52 (t, *J* = 7.8 Hz, 1H), 7.47 (s, 1H), 7.34 (d, *J* = 7.8 Hz, 2H), 7.13 (d, *J* = 8.3 Hz, 1H), 4.13 (s, 3H), 2.40 (s, 3H). ^13^C NMR (100 MHz, DMSO-*d*_6_) *δ* 155.02, 147.63, 144.82, 140.83, 138.48, 137.78, 132.57, 130.07, 129.59, 129.42 (2C), 125.76 (2C), 124.13, 123.06, 122.36, 121.88, 121.32, 116.33, 112.41, 111.34, 99.99, 35.14, 21.41. MS-ESI *m*/*z*: calcd for C_23_H_18_ClN_3_O_2_S[M + H]^+^: 436.0804; found: 436.1940.

##### 4-(tert-butyl)-N-(8-chloro-5-methyl-5H-indolo[2,3-b]quinolin-11-yl)benzenesulfonamide (**D6**)

Yield, 61%; white solid, m.p. > 400 °C (DMSO); ^1^H NMR (400 MHz, DMSO-*d*_6_) *δ* 12.73 (s, 1H), 9.01 (d, *J* = 9.9 Hz, 1H), 8.06 (d, *J* = 8.3 Hz, 1H), 8.02 (d, *J* = 8.7 Hz, 1H), 7.91 (t, *J* = 7.8 Hz, 1H), 7.80 (d, *J* = 8.1 Hz, 2H), 7.57–7.53 (m, 3H), 7.51 (s, 1H), 7.09 (d, *J* = 8.4 Hz, 1H), 4.16 (s, 3H), 1.33 (s, 9H). ^13^C NMR (100 MHz, DMSO-*d*_6_) *δ* 156.71, 148.30, 144.72, 141.53, 138.53, 137.83, 132.58, 130.04, 129.60, 125.74 (2C), 125.53 (2C), 124.22, 123.05, 122.44, 121.60, 118.08, 116.34, 110.53, 99.61, 35.14, 34.73, 31.46 (3C). MS-ESI *m*/*z*: calcd for C_26_H_24_ClN_3_O_2_S[M + H]^+^: 478.1278; found: 478.2500.

### 3.3. Cell Culture

The human gastric cell lines AGS was obtained from the American Type Culture Collection (ATCC), AGS(LOT:70012225). The human gastric cancer cell lines HGC27, MKN45, SGC7901, MGC803, liver cancer cell SMMC7721, and gastric mucosa GES-1 were obtained from the genetic resource reserve of our laboratory. The AGS, HGC27, MKN45, and SMMC7721 cells were cultured in RPMI medium (Solarbio Invitrogen Corp., Beijing, China), and the SGC7901, MGC803, and GES-1 cells were cultured in a DMEM (high glucose) medium (Solarbio Invitrogen Corp., Beijing, China) containing 10% heat-inactivated fetal bovine serum and 1% penicillin/streptomycin at 37 °C in humidified atmosphere of 5% CO_2_ in air.

### 3.4. MTT Assay

The cells were seeded in 96 well plates at the density of 8000 per well. Then, the cells were incubated in the cell incubator for about 24 h. The experiment included the treatment group, negative DMSO group, and positive group. Each group was set with 4–8 repeated wells. After the cells were incubated in the cell incubator for the corresponding time, 10 μL MTT solution of 5 mg/mL was added. The plates were incubated in the cell incubator for another 4 h. The formazan was solved in DMSO, and the absorbance value at 490 nm was measured with a microplate reader.

### 3.5. Colony Formation Assay

When AGS and HGC27 cells were in the logarithmic growth phase, AGS and HGC27 cells were inoculated into 24-well plates at a density of 1000 cells/well, and incubated in a cell incubator for 24 h. Cells were treated with compound **C5** or **C8** for 8–10 days. The cells were washed twice with PBS buffer, fixed in 4% paraformaldehyde solution for 40 min, and then stained with 1% crystal violet solution for 20 min. After staining, the cells were washed twice with fresh PBS buffer solution. The photos were taken, and Image J (National Institutes of Health, Bethesda, MD, USA) was used for counting analysis.

### 3.6. Migration Assay

A 600 μL complete medium, containing 20% fetal bovine serum, was added to the lower chamber of the transwell chamber. The cells were counted, and compounds with different concentrations were prepared using 1% complete medium. There was 150 μL cells suspension added into the upper chamber of transwell with a cell density of 50,000 cells/well. The suspension was placed in a cell incubator for further incubation for 48 h. The cells were washed twice with PBS and were fixed with 4% paraformaldehyde solution for 40 min, followed by staining with 1% crystal violet solution for 20 min. The cells in the upper transwell cells were carefully erased with cotton swabs. The number of migrating cells of the lower compartment were observed under a microscope and analyzed by Image J counting.

### 3.7. Cell Cycle Analysis

The AGS or HGC27 cells were seeded in six-well plates at a density of 800,000 cells/well. The plates were incubated in a cell incubator for about 24 h and treated with different concentrations of compounds. After 24 h, cells were collected and fixed with 1 mL of precooled 70% anhydrous ethanol for overnight. Additionally, 100 μL RNase A were added and incubated at 37 °C for 30 min. Then, the cells suspension was transferred to flow tube, and 400 μL PI solution was added and incubated at 4 °C for 30 min. The cells proportion of different cell cycles were detected by A BD FACSCanto II (BD LSRFortessa, USA). Modfit is used to process and analyze data.

### 3.8. Cell Apoptosis Analysis

AGS and HGC27 cells were seeded out in six-well plates with a cell density of 700,000 cells/well and incubated in a cell incubator for 24 h. Then, the cells were treated with different concentrations for 48 h. The supernatant was collected, the cells were washed with PBS twice, and the washing solution was collected. Then, the cells were digested and collected with trypsin without EDTA, centrifuged at 1000 r/min for 5 min, the cells were washed with PBS twice, and the cells were collected. The cell density was adjusted to 1 − 5 × 10^6^ cells/mL. Then, 100 μL cell suspension was added into the flow tube, and 5 μL Annexin V/Alexa Fluor 488 solution was added. After incubation for 5 min, 10 μL PI and 400 μL PBS solution were added. A BD FACSCanto II (BD Biosciences, USA) was used for detection, and Flowjo was used for statistical analysis.

### 3.9. Molecular Docking Analysis

Schrödinger 10.1 software (Schrödinger, USA) was used for molecular docking correlation analysis. The crystal structure of AKT protein binding to the molecule during docking was obtained from the PDB database (PDB: 6hhf). In this experiment, we used the Prime module in the Schrödinger software to fill in the missing side chains and loops in the protein, processed the protein by protonation, dehydration, hydrogenation, etc., and then minimized and initially optimized the protein under the OPLS2005 force field. The compound was then docked with the protein AKT appropriately. During the docking process, the small molecular center bound to the protein was used as the docking site of the compound to complete the whole docking process. The crystal structures of all proteins bound to molecules in docking were obtained from the PDB database, and their PDB ID, resolution, and sources could be obtained in Appendix A.

### 3.10. Western Blotting Assay

When AGS or HGC27 cells grew to about 80–90%, they were inoculated in six-well plates at a density of 500,000 cells/well and, then, incubated in cell incubator for 24 h. The cells were treated with a certain concentration of compounds for 48 h and DMSO was used as a negative control group. Cells were lysed with a high-potency tissue cell lysate containing 1% protease inhibitor PMSF and phosphatase inhibitor RIPA on ice for 30 min. Cell suspension was suspended every 10 min. After centrifugation, the supernatant was collected, and 5× protein denaturation buffer was added. The supernatant was placed in a constant temperature dry metal bath at 100 °C for denaturation for 10 min. The supernatant was stored at −20 °C. The protein was isolated with 10% separation gel and 5% concentrate gel. The total protein content was 25 μg per well. After electrophoresis, the protein was transferred to 0.22 μm PVDF membrane and blocked on a 5% skim milk shaker for 2 h. Then, the PVDF membrane was washed three times on the shaker with TBST buffer solution, 10 min each time, and blocked with primary antibody overnight at a ratio of 1:2000 at 4 °C. The PVDF membrane was washed with TBST buffer solution and incubated with a second antibody, at the ratio of 1:10,000, at room temperature for about 1 h. BCL luminescence system was used to record the photo, and the recorded protein bands were saved. Image J was used for statistical analysis of gray values.

### 3.11. Statistical Analysis

All the data were made with SPSS 22.0 (IBM, Armonk, NY, USA), GraphPad PriM 8.0 (Insightful Science, Boston, MA, USA), and AI 2020 (Adobe Systems, San Jose, CA, USA), and the error bars of all data results were represented by mean ± standard deviation. One-way ANOVA and *t*-test were used to test the differences between the analyzed data.

## 4. Conclusions

In summary, we designed and synthesized 36 neocryptolepine derivatives, based on 8-chloroneocryptolepine-based, and performed an in-depth structure-activity relationship study on five human gastric cancer cells, as well as evaluated toxicity on human normal gastric mucosa cells. Compounds **C5** and **C8** showed strong cytotoxicity to AGS and HGC27 cells. Moreover, compound **C8** inhibited the proliferation of AGS and HGC27 cells in a concentration-dependent and time-dependent manner. Cell colony formation and cell migration experiments showed that compound **C8** could inhibit the proliferation and migration of AGS and HGC27 cells, and compound **C5** could also inhibit the proliferation of AGS and HGC27 cells. Compound **C5** had a significant inhibitory effect on HGC cell migration. However, compound **C5** did not significantly affect the migration of AGS cells. In cell cycle and apoptosis experiments, the results showed that compounds **C5** and **C8** did not induce apoptosis of AGS and HGC27 cells but, mainly, caused cell necrosis. Compound **C5** had no significant effect on AGS and HGC27 cell cycle at low concentration. After treatment with 5 μM compound **C5** for 24 h, the AGS cell cycle could be significantly arrested in the G2/M phase. Compound **C8** had no significant effect on AGS and HGC27 cell cycles at 2.5 μM and 5 μM concentrations. The results of molecular docking and Western blot showed that compounds **C5** and **C8** might produce cytotoxic effects by the PI3K/AKT signaling pathway. Compounds **C5** and **C8** showed the best scores with AKT protein in molecular docking results. The experimental results showed that AKT might be the target of the compounds **C5** and **C8** acting on gastric cancer cells. In conclusion, by structural modification of neocryptolepine, two neocryptolepine derivatives with good activity were obtained, compounds **C5** and **C8**, which provided a lead compound for the development of drugs for the clinical treatment of gastric cancer.

## Data Availability

Not applicable.

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
