# Peer review of "Design, Synthesis and Biological Evaluation of Neocryptolepine Derivatives as Potential Anti-Gastric Cancer Agents"

_ijms, 2022, doi:10.3390/ijms231911924_

Round 1
Reviewer 1 Report (Previous Reviewer 3)
This manuscript describes the design, synthesis and biological testing of neocryptolepine derivatives. The methods and discussions are sound and good in most parts. There are a few statements that appear to be contradictory to the data and should have worded better. See below comments. In addition, the literature search could be a little bit more comprehensive. Overall, it is recommended this manuscript be accepted after addressing the following concerns.
1. On page 2, lines 73-76, the authors discussed the PI3K/AKT cell signaling pathway, their role in cancer development and as relevant drug targets. Actually there are a few FDA approved drugs targeting the PI3K/AKT pathway. Please refer to this paper for relevant information, Sabbah et al, Phosphatidylinositol 3-kinase (PI3K) inhibitors: a recent update on inhibitor design and clinical trials (2016–2020), Expert Opinion on Therapeutic Patents, 2021, 31:10, 877.
2. On page 2, lines 77-89. The authors reviewed current research on the neocryptolepine in terms of synthesis and activities. It seems the literature search could be more comprehensive. It is recommended that the authors include the advances in the following papers.
El-Gokha et al. Synthesis and Structure-Activity Relationships of Novel Neocryptolepine Derivatives, Chem. Res. Chin. Univ., 2017, 33(3), 373.
Emam et al. Synthesis, characterization and anticancer activity of new Schiff bases bearing neocryptolepine, J. Mol. Struct. 2017, 1146, 600.
Sidoryk et al. Searching for new derivatives of neocryptolepine: Synthesis, antiproliferative, antimicrobial and antifungal activities. Eur. J. Med. Chem., 2014, 78, 304.
3. On page 3, line 102, it says “(as shown in table 5).”. Does this Table 5 refer to Table 5 in this paper? If so, it should be labeled in order of appearance and it should be Table 1, or refer to previous publication? If latter, please cite the previous paper. It is unclear.
4. On page 3, ” Moreover, the introduction of amino long-chain alkanes at the C11 position of neopterine could improve the cytotoxic effect of the compound” there should be a reference after this sentence to support this statement.
5. On Page 4, “Chemical structure” should be “Chemical structures”
6. On Page 4, “(e) appropriate amine or hydroxyl, DMF” in Scheme 2, it is easy to understand that substitution of -Cl with amine or hydroxyl group to give amino or ether bond. However, in Table 1, there are a lot of amide moiety and sulfonamide groups, are these two groups using the same condition of (e), i.e., by reacting with amines under DMF to yield amide and sulfonamide compounds? It seems the reaction condition here is unclear.
7. Page 6, lines 175-176, “the meta-positional substitution of F atom was more 175 cytotoxic than ortho- and para-substitutions.” This statement is incorrect. In terms of IC50, the smaller the better. Thus, it is better to state “the meta-positional substitution of F atom was less potent (or less cytoxic) than ortho- and para-substitutions.”
8. Page 6, lines 177-178, “The increase of alkyl carbon chain and 177 branched chain on benzene ring had little effect on cytotoxicity” this statements seem unsupported by data. Table 4 shows the opposite, i.e., C9 and C10 with alkyl side chain shows better cytoxicity against AGS cells than compounds A1-A10.
9. Page 11, lines 315-316, “210 LEU, 263 TYR, 264 LEU, 270 VAL and 315 272 TYR”. Normally the names of the residues are before the numbers. Please rename them as LEU210 and so on.
Author Response
Dear reviewer #1:
Comment 1: On page 2, lines 73-76, the authors discussed the PI3K/AKT cell signaling pathway, their role in cancer development and as relevant drug targets. Actually there are a few FDA approved drugs targeting the PI3K/AKT pathway. Please refer to this paper for relevant information, Sabbah et al, Phosphatidylinositol 3-kinase (PI3K) inhibitors: a recent update on inhibitor design and clinical trials (2016–2020), Expert Opinion on Therapeutic Patents, 2021, 31:10, 877.
Response 1: We truly appreciate the reviewer’s comments. We have referred this paper on page 2 in revised manuscript, Sabbah et al, Phosphatidylinositol 3-kinase (PI3K) inhibitors: a recent update on inhibitor design and clinical trials (2016–2020), Expert Opinion on Therapeutic Patents, 2021, 31:10, 877.
Comment 2: On page 2, lines 77-89. The authors reviewed current research on the neocryptolepine in terms of synthesis and activities. It seems the literature search could be more comprehensive. It is recommended that the authors include the advances in the following papers.
Response 2: We appreciate the reviewer’s comments. We have added relevant literature in revised manuscript. Some studies have found that neocryptolepine derivatives have good antibacterial, anti-proliferative and antifungal activities [1-3].
Comment 3: On page 3, line 102, it says “(as shown in table 5).”. Does this Table 5 refer to Table 5 in this paper? If so, it should be labeled in order of appearance and it should be Table 1, or refer to previous publication? If latter, please cite the previous paper. It is unclear.
Response 3: Thank you for your kind suggestion. On page 3, this Table 5 refer to Table 5 in this paper. As the reviewer said, we adjusted the label in order of appearance. We have adjusted the relevant content to page 6. Further evaluation of the cytotoxicity of 8-chloroneocryptolepine on gastric cancer AGS and liver cancer SMMC7721 cells revealed that its cytotoxicity was not ideal (as shown in Table 5).
Comment 4: On page 3, ” Moreover, the introduction of amino long-chain alkanes at the C11 position of neopterine could improve the cytotoxic effect of the compound” there should be a reference after this sentence to support this statement.
Response 4: Thank you for your sincere advice. We have added a reference after this sentence to support this statement.
Comment 5: On Page 4, “Chemical structure” should be “Chemical structures”
Response 5: Special thanks to you for your good suggestion. We have revised “Chemical structure” to “Chemical structures” in the revised manuscript.
Comment 6: On Page 4, “(e) appropriate amine or hydroxyl, DMF” in Scheme 2, it is easy to understand that substitution of -Cl with amine or hydroxyl group to give amino or ether bond. However, in Table 1, there are a lot of amide moiety and sulfonamide groups, are these two groups using the same condition of (e), i.e., by reacting with amines under DMF to yield amide and sulfonamide compounds? It seems the reaction condition here is unclear.
Response 6: We acknowledge the reviewer’s comments and suggestions very much. In Table 1, we synthesized sulfonamide and amide compounds, and The same synthesis condition of Scheme 2 (e) was used for the synthesized compounds , but the statement may not be clear. We have modified our reaction conditions in Scheme 2 to “amino derivatives or hydroxyl, DMF”. This would include sulfonamides and amide compounds, making our reaction conditions more rigorous.
Comment 7: Page 6, lines 175-176, “the meta-positional substitution of F atom was more 175 cytotoxic than ortho- and para-substitutions.” This statement is incorrect. In terms of IC50, the smaller the better. Thus, it is better to state “the meta-positional substitution of F atom was less potent (or less cytoxic) than ortho- and para-substitutions.”
Response 7: Thanks to you for constructive comments and suggestions. We have modified “the meta-positional substitution of F atom was more cytotoxic than ortho- and para-substitutions.” to “ the meta-positional (D2) substitution of F atom was less potent (or less cytotoxic) than ortho- (D1) and para-substitutions (D3).”
Comment 8: Page 6, lines 177-178, “The increase of alkyl carbon chain and 177 branched chain on benzene ring had little effect on cytotoxicity” this statements seem unsupported by data. Table 4 shows the opposite, i.e., C9 and C10 with alkyl side chain shows better cytoxicity against AGS cells than compounds A1-A10.
Response 8: Thank you for your kind suggestion. We have checked the contents of Table 4 and the cytotoxicity data of compounds A1-A10 again, and as the reviewer said, we have revised the relevant content to make our presentation more accurate. The increase of alkyl side chain (C9 and C10) showed better cytotoxicity against AGS cells than compounds A1-A10.
Comment 9: Page 11, lines 315-316, “210 LEU, 263 TYR, 264 LEU, 270 VAL and 315 272 TYR”. Normally the names of the residues are before the numbers. Please rename them as LEU210 and so on.
Response 9: Thanks for your good instruction. We have revised “ 210 LEU, 263 TYR, 264 LEU, 270 VAL and 315 272 TYR” to “ LEU210, TYR263, LEU264, VAL270 and TYR272” in revised manuscript.
Special thanks to you for your valuable comments.
Reference
- El-Gokha, A.A.; Boshta, N.M.; Abo Hussein, M.K., and El Sayed, I.E.-T. Synthesis and structure-activity relationships of novel neocryptolepine derivatives. Chem Res Chin Univ. 2017, 33, 373-377.
- Emam, S.M.; El Sayed, I.E.T.; Ayad, M.I., and Hathout, H.M.R. Synthesis, characterization and anticancer activity of new Schiff bases bearing neocryptolepine. Journal of Molecular Structure. 2017, 1146, 600-619.
- Sidoryk, K.; Jaromin, A.; Edward, J.A.; Switalska, M.; Stefanska, J.; Cmoch, P.; Zagrodzka, J.; Szczepek, W.; Peczynska-Czoch, W.; Wietrzyk, J., et al. Searching for new derivatives of neocryptolepine: synthesis, antiproliferative, antimicrobial and antifungal activities. Eur J Med Chem. 2014, 78, 304-13.
Reviewer 2 Report (New Reviewer)
Ma and the co-authors reported two neocryptolepine derivatives exhibited strong cytotoxicity to AGS cells. The cell colony formation and cell migration experiments suggested that the compounds could inhibit the proliferation and cell migration of AGS and HGC27 cells. Molecular docking and Western blot showed that the compounds might induce cytotoxicity through PI3K/AKT signaling pathway. The manuscript is nicely described, and the experiments are thoughtfully designed but the reason of red colored text throughout the manuscript is not clear. I suggest accepting the manuscript after some minor modifications.
1. Compound number C5 or C8 is not in bold across the manuscript (lines 204, 250, 270, 288……)
2. Page 5 section 2.2 Please add the compound numbers when describing their cytotoxity properties like lines 144-147; replace “According to the …………………. benzene ring.” With “According ……….. the para-site substitution of F atom (A4) is better than the meta-site (A3) and ortho-site (A2) substitution, and the methoxy ortho-site (A5) substitution is more cytotoxic than the meta (A6) andpara-site (A7) substitution and the dimethoxy (A8) substitution on benzene ring”
3. Page 7 line 207-208 “In Figure 1G…… normal cells” please add the IC50 values for normal cells here.
4. Page 7, line 217-218; Delete the sentence “Moreover,….. weak” It’s a repetition of earlier sentences.
5. Figure 3. Images are not clear and impossible to ready anything out of those.
6. Figure 5. Blots are not clean and can be improved. Please try one or two necrosis marker too.
7. Figure 6. Please either move this figure after Table 5 or add few words about this figure here.
8. For M.P please mention solvent for crystallization and report melting point in the range of integral value without the decimals line 195-196 not 195.28-196.34.
9. Compound numbers under the section 3.2.5 are not in bold, please check.
10. Page 16, line 511, superscript the molecular formula.
11. Page 23, section 3.10 Mention about control (what was used as control! DMSO or what!) and please replace sealed with blocked.
Author Response
Dear reviewer #2:
Comment 1: Compound number C5 or C8 is not in bold across the manuscript (lines 204, 250, 270, 288……)
Response 1: We appreciate the reviewer’s suggestion. We have checked our manuscript and have bolded the numbers of compounds across the manuscript.
Comment 2: Page 5 section 2.2 Please add the compound numbers when describing their cytotoxity properties like lines 144-147; replace “According to the …………………. benzene ring.” With “According ……….. the para-site substitution of F atom (A4) is better than the meta-site (A3) and ortho-site (A2) substitution, and the methoxy ortho-site (A5) substitution is more cytotoxic than the meta (A6) andpara-site (A7) substitution and the dimethoxy (A8) substitution on benzene ring”
Response 2: We acknowledge the reviewer’s comments and suggestions very much. According to the reviewer's suggestion, we have added the compound numbers when describing their cytotoxicity properties in revised manuscript.
Comment 3: Page 7 line 207-208 “In Figure 1G…… normal cells” please add the IC50 values for normal cells here.
Response 3: Thank you for your guidance. We have added the IC50 values for normal cells on page 7 in revised manuscript. In Figure 2G, it was found that the cytotoxic effects of compounds C5 and C8 to normal cells (IC50 = 12.8 and 12.6 μM, respectively) were relatively weaker than gastric cancer cells.
Comment 4: Page 7, line 217-218; Delete the sentence “Moreover,….. weak” It’s a repetition of earlier sentences.
Response 4: Thanks for your good instruction. We have deleted the sentence “Moreover, the cytotoxicity of compounds C5 and C8 to normal cells was weak.”.
Comment 5: Figure 3. Images are not clear and impossible to ready anything out of those.
Response 5: Special thanks to you for your good suggestion. We have replaced the image with higher definition in revised manuscript.
Comment 6: Figure 5. Blots are not clean and can be improved. Please try one or two necrosis marker too.
Response 6: According to your kind suggestions, we have improved the definition of the photos on page 13 in the revised manuscript. The conclusions in the manuscript can be further demonstrated by adding 1-2 necrosis markers for western blot verification experiments. However, we believe that the flow cytometry results have been able to account for the cell necrosis. In our flow cytometry experiments, we used Annexin V and PI double staining method, which was able to distinguish cell apoptosis or cell necrosis.
Comment 7: Figure 6. Please either move this figure after Table 5 or add few words about this figure here.
Response 7: Thanks for your kind suggestion. According to your suggestion, we have moved Figure 6 after Table 5 and updated the order of Figures in the manuscript accordingly.
Comment 8: For M.P please mention solvent for crystallization and report melting point in the range of integral value without the decimals line 195-196 not 195.28-196.34.
Response 8: Thanks for your comments. We have added the crystalline solvent DMSO after the melting point, and we have modified the melting point of the compound to an integer value in revised manuscript.
Comment 9: Compound numbers under the section 3.2.5 are not in bold, please check.
Response 9: Special thanks to you for your good suggestion. We have revised the number of compounds in section 3.2.5 in bold and in red in the revised manuscript.
Comment 10: Page 16, line 511, superscript the molecular formula.
Response 10: Thanks to you for kind suggestions. We have subscripted the numbers in the molecular formula on page 16 according to your suggestion.
Comment 11: Page 23, section 3.10 Mention about control (what was used as control! DMSO or what!) and please replace sealed with blocked.
Response 11: Thanks to you for constructive comments and suggestions. On page 23 of the revised manuscript, Section 3.10, we have added the relevant statement, and we have checked other parts of our manuscript, the control was modified to DMSO as a negative control. We have replaced “sealed” with blocked.
Special thanks to you for your valuable comments.
This manuscript is a resubmission of an earlier submission. The following is a list of the peer review reports and author responses from that submission.
Round 1
Reviewer 1 Report
A series of neocryptolepine derivatives were synthesized by the modification of the structure of neocryptole-15 pine, and their cytotoxicity was evaluated. Compounds C5 and C8 exhibited strong cytotoxicity to gastric cancer cells AGS, which might be related to the inhibition of PI3K/AKT pathway. This work may provide lead compounds for the treatment of gastric cancer. However, the authors did not show the novelty of the work. Besides, how inhibition of PI3K/AKT pathway results in cytotoxicity and why the compounds displayed selectivity on gastric cancer cells were not illustrated. Moreover, the manuscript has not been well prepared, and following are some examples that the authors did not pay attention to:
Line 77-82, the authors did not give a reasonable explanation on the design of the target compound. Why the modification of C11 position of 8-chloroneocryptolepine was performed? How did the authors expect this modification to improve the anticancer activity?
Line 125, “Based on the above structural modification and cytotoxicity, the C11 position of 8-chloroneocryptolepine was substituted by hydrazine group to obtain compound B1-B9” is not convincing, as the authors did not explain clearly how they used the hydrazine group based on the above structural modification.
Line 278 and 279, “A series of proteins and 278 compound C5 and C8 were docked by molecular docking experiments.” Which proteins have been investigated? Why these proteins were considered?
Scheme 1 and Figure 6 were not even mentioned in the text.
Line 110, the table did not display the synthesis of compounds. So, “Synthesis and” should be deleted.
Language should be improved. For instance, in line 19 and 235, “compound” should be replaced with “compounds”; in line 121, “cyttoxicity” should be “cytotoxicity”, et al.
Reviewer 2 Report
Authors:
Your manuscript on designing, preparation and biological evaluation of neocryptolepine derivatives represent a well planned investigation. The compounds and their synthesis are well decrived. All compounds are characterized by the relevant analytical methods. The disadvantage in the biological part of this manuscript consists in a fact that no reference compound is mentioned in the Tables to be able to compare the real impact of the newly synthesized compounds. I recommend to add such a data.
Formal errors:
Latin words used in the text should be written in italics (part 2.2.). Relative configuration descriptors should also be written in italics (part 2.2.).
Figure 6: The color texts in this Figure should be checked by the English native speaker, preferably a chemist.
Summary:
I recommend to accept this manuscript after additional revision of it to complete the published data and to correct typing ot languare errors present in the currect manuscript.
Reviewer 3 Report
The authors have designed, synthesized a few neocryptolepine derivatives and tested their biological activities in vitro. There are some concerns that need to be addressed before this manuscript can be considered for publication.
1 Page 2, “strong cytotoxicity to leukemia cells MV4-11, with an IC50 of 42 nM for MV4-11 cells, and 55 also to lung cancer cells with an IC50 of 197 Nm [17].”, please delete “for MV4-11 cells” after 42 nM, and change “197 Nm” to “197 nM”
2. Page 2, a reference is needed after the following statement, “Neocryptolepine and its derivatives 56 have a wide range of biological activities, and compounds containing this ring system 57 have antifungal, antibacterial, antiviral, cytotoxic activity.”
3. Page 3, scheme 2, percert yield? The last structure missed a R1 on the benzene ring.
CH2Cl2 should be CH2Cl2 POCl3 sbould be POCl3
4. Page 4, “was substituted ether 117 group.” Should be “was substituted with ether groups.”
5. Page 4, “the compounds had a strong cytotoxicity on AGS 119 cells” should be “some of the compounds had a strong cytotoxicity on AGS 119 cells” (not all were strong, some > 50 μM)
6. Page 4, “The cytotoxic results showed that the modification of F, Cl or methoxy at benzene 120 improved the cyttoxicity. Moreover, according to the cytotoxicity results, the para-site 121 substitution of F atom is better than the ortho-site and intersite substitution, and the meth- 122 oxy ortho-site substitution is more cytotoxic than the intersite and para-site substitution 123 and the dimethoxy substitution on benzene ring” this statement should be stated with care. “
“the modification of F, Cl or methoxy at benzene 120 improved the cytotoxicity”, this statement is not necessarily true, because for A1-A10 series, A5 and A9 were the strongest, it had 2-OCH3, and 4-CH3, did not contain -F, or -Cl. And what is the “intersite” substitution referring to? For a benzene, people normally say “ortho”, “meta” or “para”, not “intersite”.
7. Page 4, why a hydrazine is introduced to substitute the 8-Cl? Any reason?
8. Page 5, for B1-B9 series, the authors stated that “and 130 ortho-substituted F atom was more cytotoxic than para- and intersite F atom.”, this is incorrect, because among 2-F, 3-F and 4-F, the IC5o for SMMC7721 shows 3-F is better than 2-F than 4-F; for AGS, 2-F is more potent than 3-F, than 4-F. Please note, for IC50, the smaller, the more potent. This statement “and 130 ortho-substituted F atom was more cytotoxic than para- and intersite F atom.” Would suggest that the larger the IC50 the better, which is just the opposite.
9. Page 5, “para-methoxide” should be “para-methoxy”
10. Page 5, “The para-methyl substitution in benzene 147 ring was more cytotoxic than ortho-methyl substitution.” This statement is incorrect, just the opposite, Table 5, D5 (AGS) IC50 is less than that of D5 (D4 (o-Me); D5 (p-Me)).
11. Page 7, Figure 1, the bond between N and C9 at H panel was not shown. Please make it appear.
12. Page 10, A series of proteins and 278 compound C5 and C8 were docked by molecular docking experiments,” both compound C5 and C8 had the best scores with AKT protein. Its 280 scores with AKT protein were -8.529 and -8.359, respectively”, what’s the unit after -9.529 and -8.359?
13. Page 10, “Aspartic acid 292 (ASP), as an acidic amino acid, will lose H in the 283 solvent, thus negatively interacting with compound C5 to form a strong negative interac-284 tion.” What does it mean to be “a strong negative interaction”? Page 10 and Page 11, “The amino groups of Lysine 268 (LYS) and arginine 273 positively interact with compound C8 in the solvent, and the Cl atoms of compound C8 molecule and the methyl 291 groups of -OCH3 are exposed to the solvent” The structures of C5 and C8 did not have a -COOH, or a -NH2, thus it is very hard to imagine they would interact with Asp292, Lys268 and Arg273, Both A and D of Figure 4 did not show such electrostatic interactions.
14. Figure 6 , SAR of page 12, because the conclusions on the SAR, see above comments, were incorrect, thus the conclusion in the box in Fig. 6 will need to be revised accordingly.
15. Page 22, “Water of crystallization, side chains, and hydrogen 778 atoms were first removed from the protein structure to minimize the energy of the entire 779 structure.” This did not sound right. Why someone would remove side chains and hydrogen atoms to minimize the protein? The selection of 6HHF as the target protein for docking is also problematic. The bound ligand was covalently bound and an allosteric binder, and orthosteric binder. For the docking studies, there are some missing residues on the AKT protein (6HHF), have the authors fixed the missing residues? If so, what and how did the authors fixed the missing residues? Similarly, the authors listed 25 proteins to dock C5 and C8. Some of those proteins contain missing residues as well, Did the authors fix the missing residues before running any docking? What is the scoring function?